# Mitochondrial Neurodegeneration

**DOI:** 10.3390/cells11040637

**Published:** 2022-02-11

**Authors:** Massimo Zeviani, Carlo Viscomi

**Affiliations:** 1Department of Neurosciences, University of Padova, Via Giustiniani 2, 35128 Padova, Italy; 2Laboratory of Mitochondrial Medicine, Veneto Institute of Molecular Medicine, Via Orus 2, 35128 Padova, Italy; 3Department of Bomedical Sciences, University of Padova, Via Ugo Bassi 58b, 35128 Padova, Italy

**Keywords:** mitochondrial disease, mitochondrial respiratory chain, OXPHOS, Leigh syndrome, MELAS, MERRF, POLG

## Abstract

Mitochondria are cytoplasmic organelles, which generate energy as heat and ATP, the universal energy currency of the cell. This process is carried out by coupling electron stripping through oxidation of nutrient substrates with the formation of a proton-based electrochemical gradient across the inner mitochondrial membrane. Controlled dissipation of the gradient can lead to production of heat as well as ATP, via ADP phosphorylation. This process is known as oxidative phosphorylation, and is carried out by four multiheteromeric complexes (from I to IV) of the mitochondrial respiratory chain, carrying out the electron flow whose energy is stored as a proton-based electrochemical gradient. This gradient sustains a second reaction, operated by the mitochondrial ATP synthase, or complex V, which condensates ADP and Pi into ATP. Four complexes (CI, CIII, CIV, and CV) are composed of proteins encoded by genes present in two separate compartments: the nuclear genome and a small circular DNA found in mitochondria themselves, and are termed mitochondrial DNA (mtDNA). Mutations striking either genome can lead to mitochondrial impairment, determining infantile, childhood or adult neurodegeneration. Mitochondrial disorders are complex neurological syndromes, and are often part of a multisystem disorder. In this paper, we divide the diseases into those caused by mtDNA defects and those that are due to mutations involving nuclear genes; from a clinical point of view, we discuss pediatric disorders in comparison to juvenile or adult-onset conditions. The complementary genetic contributions controlling organellar function and the complexity of the biochemical pathways present in the mitochondria justify the extreme genetic and phenotypic heterogeneity of this new area of inborn errors of metabolism known as ‘mitochondrial medicine’.

## 1. Introduction

### 1.1. Mitochondrial Biology

Mitochondria are double-membrane cytoplasmic organelles playing a central role in energy metabolism and several other metabolic and execution pathways. The mitochondrial respiratory chain, MRC, is the final common pathway for the aerobic synthesis of ATP via oxidative phosphorylation (OXPHOS). Structurally, OXPHOS is carried out by five multiheteromeric MRC enzymatic complexes, embedded in the inner mitochondrial membrane, IMM, and two electron shuttles, coenzyme Q (CoQ, operating electron translocation from complex I (CI) and CII to CIII), and cytochrome c (cyt c, translocating electrons from CIII to CIV, or cytochrome c oxidase, COX). Several other pathways important in intermediary metabolism are found within the mitochondrial matrix, including the pyruvate dehydrogenase complex (PDc), which oxidatively converts pyruvate to acetyl-CoA, and the enzymes of fatty acid beta-oxidation, which both feed the Krebs or tricarboxylic acid (TCA) cycle and eventually the MRC. Whilst defects in all of these pathways can be defined as mitochondrial, the term ‘mitochondrial disorder’ is usually restricted to clinical syndromes associated with the significant number of metabolic, genetically determined abnormalities of the respiratory chain leading to OXPHOS failure [1].

The orchestrated actions of the MRC complexes can be subdivided into two distinct reactions [2,3]. The first reaction, called mitochondrial respiration, is carried out by the proton-translocating respiratory chain complexes I, III, and IV (CI, CIII, and CIV), and a set of other redox enzymes, including CII (succinate–ubiquinone oxidoreductase), which is also part of the TCA cycle; the electron-transfer flavoprotein–ubiquinone oxidoreductase (ETF-QO); and dihydroorotate dehydrogenase, a key enzyme for the de novo biosynthesis of pyrimidines. Respiration operates the flow of electrons stripped off the nutrient molecules oxidatively catabolized in the TCA cycle and, for free fatty acids, also by the beta-oxidation spiral, with the ultimate reduction of molecular oxygen to water, a terminal step performed by COX. In three steps of respiration, those involving CI, CIII, and CIV, the electron flow sustains the action of proton pumps that translocate protons from the inner to the outer mitochondrial compartments, across the IMM. This proton translocation leads to the formation of a mitochondrial membrane potential, MtMP, composed of a chemical gradient, ΔpH, and an electrostatic gradient, ΔΨ, which provide the electron-motive force, Δp, exploited by the H^+^-dependent ATP synthase (CV), to carry out the secΨond OXPHOS reaction, i.e., the rotational-based condensation of ADP+Pi into ATP. This is made possible by protons that propel the CV rotor by crossing the IMM back to the inner mitochondrial compartment, by traversing an oblique transmembrane channel largely composed by subunit A of CV, encoded by mt-ATPase6, a mitochondrial DNA (mtDNA)-encoded gene. Human mtDNA is a 16.5 kb circular, double stranded DNA, encoding only 13 of the approximate 85 polypeptides included in the OXPHOS complexes (seven subunits of CI, one of CIII, three of CIV, and two of CV), plus two rRNAs and 22 tRNAs, i.e., the RNA apparatus required for the largely autonomous mitochondrial translational machineries [4,5]. All of the other mitochondrial proteins, including those required for mtDNA maintenance and expression, OXPHOS assembly, and biosynthesis of cofactors, such as hemes and iron–sulphur (Fe–S) redox clusters, are nuclear-coded and imported to mitochondria.

Mitochondrial DNA differs from nuclear DNA in several ways, the most important being its strict maternal inheritance in sexuate organisms, and, in all eukaryotes, its presence in multiple copies, from thousands to hundreds of thousands, within each cell of the body (polyplasmy). Under normal conditions, all copies of mtDNA in an individual are identical, a situation known as homoplasmy. Coexistence of more than one population of mtDNA, i.e., wild-type (wt) mtDNA and one or more mutant mtDNA species, is called heteroplasmy and is a frequent feature of mtDNA-related disorders. The combination of uniparental inheritance and presence of multiple copies in every cell has major implications for our understanding of genetically determined mitochondrial disorders. Unlike chromosomal genes, where classical Mendelian patterns of inheritance are, in the majority of cases, dictated by the presence of two alleles (diploid gene organization), mutations of mtDNA can affect a continuous proportion of copies, from 0% to 100%. Mitochondrial DNA is also highly polymorphic, with an estimated mutation rate of 7–10 times higher than that of nuclear DNA [6]. Mutations must therefore arise continuously and are either fixated as ‘neutral’ polymorphisms, eventually eliminated or prevailing, by natural selection, or never reach a level at which cellular dysfunction ensues. However, the amount of mutant mtDNA must rise over a threshold level before a cellular phenotype is manifested. At this level, the mutation will no longer be complemented by the coexisting wt-mtDNA. Therefore, phenotypic expression depends on the remaining amount of wt-mtDNA, together with its intrinsic pathogenicity, its tissue distribution, and the relative reliance of each organ system on the mitochondrial energy supply. Studies in human families have shown that there can be dramatic changes in the level of heteroplasmy from one generation to the next. This has led to the suggestion that there is a reduction or constriction of the amount of genetic information (i.e., number of mtDNA copies) flowing from one generation to the next, the so-called ‘bottleneck hypothesis’ [7,8]. This bottleneck occurs in the female germline. The genetic information (segregating units) that becomes available to the next generation consists of individual mtDNA molecules physically organized in transmissible structures called mitochondrial nucleoids. It is now widely accepted that each nucleoid contains, on average, just a single mtDNA molecule [9]. The number of these segregating units has been estimated to be as few as 200 in mice and possibly even less in humans. In addition to the rapid changes in heteroplasmy that arise from the bottleneck, stochastic distribution of mtDNA to subsequent cells during mitosis can also lead to major differences in the proportion of mutant and wt-mitochondrial genomes in cells and tissues. This phenomenon, called mitotic segregation, may contribute to the extreme phenotypic variability of any given mtDNA mutation that is often observed in mitochondrial disorders.

While the gene products encoded by mtDNA are essential, they comprise only a fraction of the proteins involved in a functional MRC. Nuclear genes are encoding the greater number of MRC subunits and are also required for the transport of proteins to the mitochondrion, uptake and assembly, and many other functions necessary to build and maintain a functional OXPHOS system [1]. In addition, mtDNA replication, transcription, and translation are absolutely dependent on nuclear gene products [10]. Importantly, mitochondria are highly dynamic organelles capable of organizing themselves, under specific metabolic conditions and in specific tissues, as an interconnected tubular network or fragmented array of individual organelles, by fission and fusion events [11]. A complex protein apparatus is involved in mitochondrial network dynamics, as well as in regulating mitochondrial morphology, cristae organization, maintenance of membrane potential, autophagy of portions, or whole energetically spent or abnormal mitochondria, and the compartmentalization of proapoptic factors.

### 1.2. Mitochondrial Pathophysiology

As shown in Figure 1, inherited defects of OXPHOS comprise mutations in mtDNA or in OXPHOS-related nuclear genes [1].

### 1.3. Mutations in Genes Encoding MRC Subunits

Table 1, Table 2, Table 3, Table 4 and Table 5 display the current knowledge about structural and accessory genes involved in mitochondrial disorders associated with isolated defects of CI, CII, CIII, CIV, and CV. Neurological failure, either alone or in combination with other organ abnormalities, is present in the vast majority of the cases.

With 45 subunits, CI is the first and largest complex of the respiratory chain. It is under dual genomic control; a proper interaction between the mitochondrial and the nuclear genome is clearly important for proper biogenesis and functioning of the complex. Isolated CI deficiency is the most frequently diagnosed form of mitochondrial disorders caused by the disturbance of the OXPHOS system. CI deficiency has a wide clinical variety, and the underlying genetic cause of CI deficiency is still not known in many patients. Importantly, several conditions with CI deficiency are caused by mutations in assembly factors involved in the formation of the complex. One of these, DNAJC30, has been recently associated with an autosomal recessive phenocopy of LHON, but also with a couple of cases of recessive Leigh disease (LD) [15,16]. The spectrum of neurological presentations in CI defects [17] includes LD [18,19], progressive leukodystrophy [20], and severe neonatal lactic acidosis. The molecular dissection of the structural components of CI in LD is still ongoing.

Disease-causing mutations have been described in the genes encoding CI assembly factors and/or cell biological studies have shown the involvement of these proteins in CI formation. On the contrary, the role of the putative assembly factors (indicated with the term ‘Assembly?’ in Table 1) needs to be established. As indicated, NDUFA4 has unequivocally been recently attributed to CIV (cytochrome c oxidase) [13].

Mitochondrial CII (succinate–ubiquinone oxidoreductase) is the smallest complex of the OXPHOS system, a tetramer of just 140 kDa. Despite its diminutive size, it is a key complex in two coupled metabolic pathways: it oxidizes succinate to fumarate in the TCA cycle and the electrons are used to reduce FAD to FADH_2_, ultimately reducing ubiquinone to ubiquinol in the MRC. The biogenesis and assembly of CII is facilitated by some ancillary proteins, all of which are autosomally-encoded. Numerous pathogenic defects have been reported, which describe two broad clinical manifestations, either susceptibility to neoplasia/cancer in the case of single, heterozygous germline variants, or a mitochondrial disease presentation, almost exclusively due to bi-allelic recessive variants associated with an isolated CII deficiency. After the first report concerning two children with an LD phenotype and a point mutation in the gene encoding the flavoprotein (Fp) subunit of succinate dehydrogenase [22], approximately 60 patients have been described with CII deficiency, harboring more than 30 different pathogenic variants in the four structural CII genes, encoding subunit genes (SDHA, SDHB, SDHC, and SDHD) and genes encoding two CII assembly factors (SDHAF1 and SDHAF2). There is phenotypic heterogeneity associated with defects in each CII gene, similar to other mitochondrial diseases. Inborn errors of CII proteins causing MRC dysfunction are rare. As mentioned above, mutations in the subunits of CII, as well as in the assembly factor SDHAF2, have been identified in patients with different forms of neoplasia, particularly familial paraganglioma and pheochromocytoma, rare neural crest tumors (reviewed in [23]).

Within the group of mitochondrial diseases, or more specifically, OXPHOS disorders, isolated mitochondrial CIII deficiencies are among the least frequently diagnosed. It is possible that these deficits are not rarer than those of the other complexes, but their diagnosis may be more difficult due to the lack of histological and biochemical hallmarks in skeletal muscle biopsies, e.g., no COX negative fibers [25]. Moreover, different protocols used in different labs to measure CIII enzymatic activity can introduce some bias to detect defects [26]. Typical to mitochondrial syndromes, CIII defects are associated with a wide range of clinical presentations, the only common feature being the reduced ubiquinol–cytochrome c oxidoreductase enzymatic activity measured in biological samples of the patients. The defective factor responsible for CIII malfunction and thus, the molecular pathogenic mechanisms are also widely variable. As shown in Table 3, some CIII-associated disorders are caused by mutations in structural CIII subunits, including the mtDNA-encoded cytochrome b, as well as some nucleus-encoded structural gene products. In several cases, the responsible genes encode CIII assembly factors, including BCS1L, essential for the incorporation of UQFRS1, the Rieske Fe–S protein, as the terminal step for CIII formation. Mutations in BCS1L on chromosome 2q35 (OMIM 603647) are the most frequent cause of mitochondrial CIII isolated deficiency [27,28,29,30]. BCS1L mutations are associated with a wide variety of clinical manifestations with different tissue involvement and disease progression, ranging from multivisceral GRACILE syndrome (growth retardation, aminoaciduria, cholestasis, iron overload, lactic acidosis, and early death; OMIM 603358; [31,32,33]); to congenital metabolic acidosis, neonatal proximal tubulopathy, and/or liver failure with or without encephalopathy [34,35,36,37,38,39]; to isolated severe mitochondrial encephalopathy [29]; to milder phenotypes, such as Björnstad syndrome (sensorineural hearing loss and pili torti; OMIM 262000; [40,41], a neurological syndrome with long-term survival [42], or neuro–psychiatric manifestations [43]. Additionally associated with the handling of UQCRFS1 is human LYRM7, located on chromosome 5q23.3 (OMIM 615831), which encodes the LYR (leucine/tyrosine/arginine)-motif protein 7, a member of the CI_LYR-like superfamily [44]. LYR motifs are the molecular signature of proteins that contain or assist in the delivery of Fe–S clusters. In contrast with BCS1L and LYRM7 (also termed MZM1L) which both have orthologues in yeast, a third assembly factor, TTC19, has been found only in animals but neither in plants nor yeasts [45]. TTC19 (OMIM 613814), located on chromosome 17p12, encodes the tetratricopeptide repeat domain-containing protein 19 (TTC19) involved in CIII biogenesis. The first cases of CIII deficiency associated with TTC19 mutations were described in three patients from two unrelated Italian families with early-onset but slowly progressive encephalopathy, and in a fourth patient with late-onset but rapidly progressive neurological failure [45]. Since the first cases were reported, other TTC19 mutations have always been associated with isolated CIII but with different clinical presentations. A progressive neurodegenerative disorder showing severe psychiatric signs and cerebellar disease was found in four Portuguese siblings born to consanguineous parents [46], LD was reported in an Hispanic child [47], and cerebellar ataxia has been reported in Japanese adult individuals [48,49]. All the described cases carried non-sense or frameshift mutations leading to a truncated protein and, at least in the first reported patient samples, to undetectable TTC19 levels [45]. However, the clinical output may vary with age of onset, severity, and presence of psychiatric symptoms. TTC19 was shown to co-immunoprecipitate and co-migrate in blue-native gel electrophoresis (BNGE) with several CIII structural subunits, suggesting physical interaction. Although its exact function is currently unknown, a role as a chaperone in the first steps of CIII assembly is proposed for this protein, because a proportion of unassembled UQCRC1 and UQCRC2 was found in mutant muscle samples [45,50].

Cytochrome c oxidase (COX) deficiency is characterized by a high degree of genetic and phenotypic heterogeneity, partly reflecting the extreme structural complexity, multiple post-translational modification, variable, tissue-specific composition, and the high number of and intricate connections among the assembly factors of this enzyme. In fact, decreased COX-specific activity can manifest with different degrees of severity, affect the whole organism or individual tissues, and develop a wide spectrum of natural history, including disease onsets ranging from birth to late adulthood. More than 30 genes have been linked to COX deficiency, but the list is still incomplete, despite being constantly updated. Interestingly, mutations of COX structural genes, either the three mtDNA genes encoding core subunits, or the 11 or more (including tissue specific isoforms) encoded by nuclear genes, are extremely rare. On the contrary, a significant number of COX-specific assembly factors are responsible for most of the COX-defective syndromes, probably including the single most common cause of LD, namely mutations in the COX chaperone Surf1 (see below).

Under physiological conditions, the mitochondrial ATP synthase provides most of the energy to the cell via OXPHOS. Alterations of OXPHOS mainly affect the tissues characterized by a high-energy metabolism, such as nervous, cardiac, and skeletal muscle tissues. The most frequent mutations in CV are probably those affecting the mitochondrial ATPase6 gene, encoding subunit A of the enzyme. Again, an important disease gene associated with profound ATP synthase deficiency encodes a putative assembly factor, TMEM70. Interestingly, TMEM70 seems to also play a role in the formation-stabilization of CI, in addition to a well-established function in the assembly of CV. Nd stands for not determined.

### 1.4. Mitochondrial DNA Mutations

A significant number of reports on pathogenic mutations of mtDNA have been accumulating in the last three decades, in association with a wide spectrum of clinical presentations [1,53,54,55]. Despite the impressive number of disease-related mutations identified in recent years [56], new pathogenic mutations continue to be reported (http://www.mitomap.org) (las access 20 December 2021). For instance, a constellation of mutations in the seven ND genes encoded by mtDNA is responsible for a substantial fraction of isolated defects of CI [56]. The flood of ‘novel’ mtDNA mutations that inundates the neurological literature, however, has prompted population geneticists to cast doubts on the accuracy of previously published studies, particularly on the criteria adopted to define the pathogenic role of novel mtDNA variants [53]. On the other hand, because of the extreme complexity of OXPHOS and its peculiar genetic organization, the number of (nucleus-encoded) genes potentially involved in disease is enormous and tends to coincide with the size of the mitochondrial proteome itself, which is estimated to total nearly 1500 gene products.

### 1.5. Syndromes Associated with mtDNA Instability

An important group of diseases comprises Mendelian traits characterized by either the accumulation of multiple mtDNA deletions or the loss of mtDNA in affected tissues [55].

Mutations in pol-γ, the only mtDNA-specific polymerase, have been associated to both types of mtDNA instability [57]. The pol-γ holoenzyme consists of a single 145 kDa catalytic subunit (pol-γA, encoded by the POLG gene), which forms a heterotrimeric complex with two identical 55 kDa accessory subunits (pol-γB, encoded by POLG2) [10]. More than 100 mutations in pol-γA have been reported so far (see http://tools.niehs.nih.gov/polg (accessed on 1 November 2021) for a complete and continuously updated list of POLG mutations associated with different syndromes). Different mutations affecting different domains of pol-γA, including the N-terminal proofreading domain, the C-terminal catalytic polymerase domain, and an intermediate ‘spacer’ region, which binds to pol-γB, thus regulating polymerase processivity, can affect one or more of its enzymatic properties, as shown both in vitro and in model organisms [10,58]. The clinical outcomes vary [57]. Autosomal dominant or recessive progressive external ophthalmoplegia (PEO) with proximal myopathy may be complicated by neurodegenerative abnormalities in the central and peripheral nervous systems, for example, parkinsonism, and occasionally by extraneurological symptoms. A juvenile-onset recessive syndrome is characterized by spinocerebellar ataxia with epilepsy (SCAE), while infantile presentations typically occur in the form of hepatopathic poliodystrophy (Alpers–Huttenlocher syndrome, AHS). SCAE and AHS are often associated with the segregation of alleles carrying two specific mutations in the pol-γA spacer domain, which indicates the existence of a continuum clinical spectrum of brain and liver damage, correlated with a common pathological pol-γA genotype. Dominant or recessive PEO syndromes are typically associated with the accumulation of multiple mtDNA deleted molecules in affected tissues, while AHS shows depletion of mtDNA in the liver and possibly the brain. The mtDNA lesions of SCAE are less well documented, but mtDNA depletion in brain regions has been detected.

Multiple mtDNA deletions and PEO are not exclusive of pol-γA mutations, but can also be found with mutations in several additional genes: ANT1, the muscle specific isoform of the adenine nucleotide translocator [59]; TWINKLE, the mtDNA helicase [60]; pol-γB [61], but also a wide set of genes partly involved in mtDNA metabolism. For instance, recessive mutations in RNAse H1, an enzyme digesting the RNA components of DNA-RNA hybrids, present in both the nucleus, together with RNAse H2, responsible for Aicardi–Goutieres syndrome, as well as mitochondria, where only RNAse H1 is found. The genetic transmission may be either autosomal dominant, such as for many POLG, ANT1, and c10orf2 (encoding TWINKLE) mutations, or recessive, such as for mutations in the RNAse H1 gene [62]. The predominant clinical presentation is in the extraocular and proximal muscle districts, including in some cases, the respiratory intercostal muscles and diaphragm, but also involving the central and peripheral nervous systems and other organs, as seen in PEO associated with pol-γA or RNAse H1 mutations. A specific recessive mutation in C10ORF2 (encoding TWINKLE) is associated with infantile spinocerebellar ataxia (IOSCA) [63], a neurological syndrome belonging to the Finnish disease; heritage and recessive C10ORF2 mutations may occasionally cause AHS as well [64,65]. AHS is an example of severe hepatocerebral mtDNA depletion syndrome (MDS), due to pol-γA (and rarely TWINKLE) recessive mutations, but MDS can also be caused by mutations in a number of other factors that control the mitochondrial or cytosolic supply of deoxynucleotides, the ‘building blocks’ of mtDNA. Mutations in mitochondrial deoxyguanosine kinase (dGUOK) are responsible for a hepatocerebral form of infantile MDS, in which the clinical picture is dominated by liver failure and progressive neurological lesions [66]. Mutations in two other genes encoding mitochondrial thymidine kinase 2: TK2 [67] and cytosolic P53-dependent 2B subunit of ribonucleotide reductase, RRM2B [68], are both associated with severe MDS in skeletal muscle, while defects of the ATP-dependent succinyl-CoA ligase, SUCLA2, cause multisystem, predominantly encephalopathic syndromes, which combine MDS with the presence of methylmalonic acid in body fluids [69,70,71]. A peculiar form of hepatocerebral MDS is due to mutations in the MPV17 gene [72]. A specific mutation in MPV17, originally found in patients from an Italian family, was later demonstrated to segregate with Navajo neurohepatopathy (NNH) [73]. Finally, mutations in TYMP/ECFG1, the gene encoding thymidine phosphorylase, TP, an enzyme involved in the catabolism of pyrimidines, are responsible for myo-neurogastrointestinal encephalopathy (MNGIE) [74]. In MNGIE, the accumulation of thymidine determines a toxic imbalance of the nucleotide pools, leading to the instability of mtDNA in critical tissues [75]. This phenomenon has been documented experimentally in other forms of MDS [76]. Clearance of thymidine by peritoneal dialysis [77], and more recently as well as more effectively, by allogenic bone marrow stem cell [78] or liver transplantation [79], are promising rational therapies for MNGIE. The newest entry in the family of MDS genes is SSBP1, encoding the single-stranded mitochondrial DNA binding protein 1 (mtSSB1), an essential gene for mtDNA replication. Heterozygous missense mutations of SSBP1 have recently been found in optic atrophy (OA) and foveopathy associated with MDS [80,81,82,83]. In one patient blindness was followed by hypertrophic cardiomyopathy, nephropathy, ataxia, and growth retardation. Muscle biopsy revealed COX-negative fibers; biochemical studies documented a combined deficiency of CI and CIII, whereas citrate synthase (CS), an index of mitochondrial mass, was elevated. In cases where the mtDNA copy number was measured, MDS was found in skeletal muscle biopsy and other specimens [80]. In fibroblasts, SSBP1 mutant patients displayed reduced mtDNA content, ranging from 54% to 78% depletion compared to controls.

### 1.6. mtDNA Translation Defects

In addition to mtDNA instability, neurological derangement can result from abnormalities in the translation of the 13 mtDNA structural genes into their corresponding proteins. The RNA apparatus (22 tRNAs and two rRNAs) serving autochthonous mitochondrial protein synthesis is encoded by mtDNA; mutations in these genes are a well-established cause of a number of maternally inherited mitochondrial syndromes, including mitochondrial encephalomyopathy, lactic acidosis and stroke-like episodes (MELAS), myoclonic epilepsy with ragged-red fibers (MERRF), and aminoglycoside-induced hearing loss [55]. A second group of clinical conditions has recently been linked to mutations in nuclear genes encoding some of the very many protein factors involved in mtDNA translation [84]. For instance, extremely severe, early-onset, fatal syndromes can be due to mutations in MRPS16 [85], encoding a protein component of the small ribosomal unit, or in each of the three mitochondrial elongation factors, EFG1 [86,87], EFTu [87], and EFTs [88]. In some cases the clinical presentation is exclusively neurological, while other patients suffer from multivisceral involvement [89]. Another interesting clinical presentation combines childhood or juvenile-onset myopathy with lactic acidosis and sideroblastic anemia (MLASA). MLASA is caused by mutations in pseudouridylate synthase (PUS)1, an isomerase that converts uridine into pseudouridine at several positions of both cytosolic and mitochondrial tRNAs [90]. Two PUS1 isoforms containing signals targeting either the nucleus or the mitochondrion, are encoded by the same gene. This regulatory mechanism along with the redundancy of the PUS enzyme family, can partly explain the heterogeneity of the clinical manifestations in MLASA. Additional syndromes, caused by mutations in mitochondrial aminoacyl-tRNA synthetases, have been identified in a number of clinically heterogeneous patients who display a spectrum of conditions, including an MLASA phenocopy due to YARS2 mutations, as well as isolated or syndromic forms of early-onset leukoencephalopathy (see below).

### 1.7. Mutations in Genes Controlling the Synthesis of Specific Mitochondrial Lipids and Cofactors

In mitochondria, ubiquinone (coenzyme Q10; CoQ10) funnels electrons to CIII. Mutations in the CoQ_10_ biosynthetic genes, *COQ2* [91,92], *PDSS1* [91], and *PDSS2* [93] cause severe infantile syndromes associated with CoQ_10_ deficiency, whereas the molecular genetic basis of adult-onset CoQ_10_ deficiency remains undefined. Low levels of CoQ_10_ have been reported in muscle biopsies of patients with ataxia-oculomotor-apraxia 1 (AOA1) due to mutations of aprataxin, supporting the hypothesis that the ataxic form is a genetically heterogeneous disease in which CoQ_10_ deficiency can be secondary [94]. Recently, mutations in the *ADCK3* gene were found to cause childhood-onset ataxia and in three of these patients CoQ_10_ levels were low [95].

The two genes encoding the enzymes COX10 and COX15, involved in the synthesis of heme a, the prosthetic group of COX, are mutated in different clinical presentations, including neurological syndromes, such as Leigh syndrome [96]. See Table 4.

A defect in the mitochondrial pathway deputed to detoxify hydrogen sulphide (H_2_S), which is ultimately oxidized to SO_4_^2+^, is responsible of ethylmalonic encephalopathy (EE), a devastating infantile brain disorder caused by the inhibitory action of H_2_S on cytochrome c oxidase [97]. The responsible gene, *ETHE1*, is a sulphur dioxygenase (OMIM *60845) containing a single Fe atom, which is recessively mutated in EE [98]. ETHE1 protein converts S_2_H into sulphite, the substrate of sulphite oxidase (see below) [97]. Interestingly, both the replacement of the crippled enzyme in the liver by AAV gene therapy in a KO mouse model, and liver transplantation in EE patients have prevented or arrested the disease progression effectively [99,100]. This is an example of a mitochondrial disease caused by the accumulation of toxic compounds, which in this case is H_2_S. Another infantile fatal condition affecting the same mitochondrial pathway is associated with mutations of *SUOX*, encoding a molybdenum-based sulphite oxidase, the terminal component of the H_2_S detoxifying pathway (OMIM *606887) [101]. Clinically, the disease can ensue very early after birth or be characterized by later onset and more prolonged survival. The syndrome is dominated by neurological failure, including ssive ataxia, stagnation/regression of neurological milestones, and seizures in the later infantile cases. Finally, recessive missense mutations in cytochrome c have been associated with thrombocytopenia, whereas mutations in the X-linked gene encoding the holocytochrome c-type synthase (HCCS) gene is lethal in males and associated with non-canonical mitochondrial dependent apoptosis in females, which explains the associated phenotype of microphthalmia with linear skin lesions (MLS) [102].

### 1.8. Variable Penetrance and Tissue Impairement in Mitochondrial Disorders

Tissue specificity may limit the systemic effect of metabolic changes, whilst still inducing marked abnormalities within the affected tissue. There are, however, likely to be many contributing factors, including different metabolic needs of a tissue, tissue-specific expression of nuclear OXPHOS genes, and tissue-dependent segregation of heteroplasmy, when present.

Homoplasmic mtDNA mutations, such as those determining Leber’s hereditary optic neuropathy (LHON), are typically associated with variably reduced penetrance. To date, LHON is the only human disease for which the influence of the mtDNA background (haplogroups) has been solidly documented [103], particularly on the m.14484T>C in *MT-ND6* and m.11778G>A in *MT-ND4* LHON mutations [104], whereas the association of LHON with a chromosome X locus is still a controversial, but interesting hypothesis since it could explain the male prevalence of this mitochondrial disorder [105].

### 1.9. Redox Abnormalities and Intrinsic Apoptotic Pathway

Besides a shortage in ATP production leading to energy failure, other pathogenic processes may include excessive production of reactive oxygen species (ROS), the release of apoptotic signals, abnormalities in calcium homeostasis [106], abnormalities in the fission/fusion, and distribution of the organelles, and other poorly defined processes. Studies carried out in yeast, experimental animals, and human cells support the idea that ROS play a relevant role at least in some mitochondrial disorders. Different mtDNA variants may significantly influence ROS generation in mice [107]. However, the relationship between ROS production, accumulation of mtDNA point mutations, and aging is still controversial, as indicated by well-established data on a pol-γA mutator mouse [108]. Finally, a study on a fly model of neurofibromatosis-1 (NF1) indicates that overexpression of neurofibromin increases lifespan through cAMP regulation of mitochondrial respiration and ROS production [109].

Although mitochondria are ubiquitous and any organ can be affected by mitochondrial abnormalities, at any age, with any clinical course, and through any kind of genetic transmission (including sporadic cases), the central nervous system (CNS) as well as the peripheral nervous system (PNS), for mutations in specific genes, are, together with skeletal muscle, the most frequent targets of mitochondrial impairment.

### 1.10. Mitochondrial Neurological Disorders

Neurological impairment is a hallmark feature of primary mitochondrial disorders. In this review, we will discuss mitochondrial neurodegenerative conditions in (a) infants and children, and (b) adults, since the clinical and genetic features may be very different in these two groups of patients. We shall not address LHON in depth, as recent reviews are available on this specific issue (e.g., Zeviani and Carelli, in press). In addition, we will not discuss the role of mitochondria in common neurodegenerative diseases, such as Parkinson’s, Alzheimer’s, and ALS. For these conditions, we refer the reader the literature [110,111,112].

## 2. Specific Neurodegenerative Syndromes in Infants and Children

### 2.1. Necrotizing Encephalomyelopathy: Leigh Disease (LD)

#### Clinical Definition

Leigh disease (LD), sometimes termed Leigh syndrome (LS), is the most common mitochondrial brain disease in infancy and childhood. It is primarily defined for its peculiar neuropathological–neuroimaging pattern [113], consisting of symmetric lesions in areas of the subcortical nuclei, including upper spinal cord metamers, brain stem nuclei, thalamus, white and grey matter in the cerebellum, and the striatum. The lesions are characterized by demyelination with reactive gliosis, necrotic areas with spongiosis, and the proliferation of microcirculation (Figure 2).

Despite the typical LD neuropathological and neuroimaging patterns are genetically heterogeneous, they should prompt clinicians toward a diagnosis of impaired mitochondrial bioenergetics. LD has in fact been recognized as the common neuropathological consequence of early failure in mitochondrial bioenergetics, the essential source of ATP allowing nerve cells and interneuronal connections to survive and function in the brain. An important laboratory hallmark of the disease is increased levels of lactic acid in both the CSF and the blood due to impaired cellular respiration. The neurological symptoms are related to the function of the neurodegenerated/necrotic CNS structures and may start with generic hypotonia, regression/stagnation of psychomotor milestones, to then evolve with variable onset of dystonia, motor hyperactivity, and incoordination (chorea and ataxia), and eventually the appearance of spastic quadriparesis associated with failure to thrive, quadriparetic spasticity, accompanied by general symptoms that include failure to thrive and overwhelming vomiting, possibly due to increased lactic acidosis. The PNS may also be involved usually as a mixed axonal-demyelinating polyneuropathy. Failure of the CNS, together with PNS and skeletal muscle insufficiency, dominate the clinical picture, although, occasional failure of the proximal renal tubule may be observed as De Toni–Debré–Fanconi syndrome. MRI findings in LS typically include the presence of bilateral lesions from the rostral spinal cord through the brainstem, including the cerebellum, to the diencephalon, up to the basal ganglia (Figure 3). Failure in the different brainstem nuclei and tracts as well as basal ganglia and cerebellar functions account for the main clinical features of LD.

### 2.2. Molecular Genetics

Severe defects in any MRC complex, either isolated or in combination, can be associated with LD (OMIM 256000). The most frequent deficits can be categorized into four groups: (i) single defects of CI or (ii) CIV; or (iii) multiple MRC defects; and (iv) mutations in *MTATP6* mutations. Only one CIII mutation determining a p.Ser45Phe homozygous replacement in *UQCRQ* encoding a small structural subunit [116] has been associated with LD. In our experience, defects of pyruvate dehydrogenase complex (PDHC) activity are relatively frequent in LD, in particular those associated with mutations in the X-linked PDHA1 gene encoding the E-1α catalytic subunit of the complex [117]. However, PDHC deficiency can also be found in a number of early-onset encephalopathy cases with neuroradiological features that differ from, or only partially overlap with, those of typical LD.

### 2.3. Complex I Defects

Defects of CI have been found frequently in LS striking mtDNA-encoded subunits (e.g., mutations in *MTND2*, *MTND3*, *MTND5*, and *MTND6*), or nucleus-encoded subunits (*NDUFS1*, *NDUFS3*, *NDUFS4*, *NDUFS7, NDUFS8, NDUFA2*, and *NDUFV1*), as well as pathogenic changes in CI-associated assembly factors (*NDUFAF2*, *C8Orf38*, *C20Orf7*, and *FOXRED1)*. See Table 1 for a more complete list. 

### 2.4. Complex IV Defects 

Mutations in genes encoding COX assembly factors are the most frequent cause of impaired CIV activity, being transmitted as an autosomal recessive trait. Early-onset LD is the usual clinical presentation. The most frequent gene responsible for COX-defective LD is encoded by *SURF1* and plays a role as an early-assembly factor during the formation of the enzyme through a still baffling mechanism [118]. However, infantile severe encephalopathy can be due to mutations in *COX10* [119] and *COX15* [120] as well as *TACO1* [121]. The first two genes encode enzymes necessary for the farnesylation (COX10) and hydroxylation (COX15) of the heme moiety eventually becoming mature functionally active heme a, whereas TACO1 is a mitochondrial factor required for efficient translation of COX subunit I. A single mutation (p.Ala354Val) in *LRPPRC* [122], encoding a mitochondrial pentatricopeptide playing a general role in mtDNA transcription, including mt-RNA stability and processing, is responsible for the French-Canadian variant of LS (OMIM 220111). See Table 4 for a full list. As observed for many other mitochondrial disease genes, in a few instances, pathological alleles in *SURF1* may be associated with a different form of neurological disease, namely a demyelinating CMT syndrome (CMT4K) with some cerebellar and other CNS signs [123].

### 2.5. MRC Combined Defects

Mutations in the mitochondrial elongation factor *EFG1* gene were identified in only one subject with early-onset LS [87]. Homozygous mutations in the c12orf65 gene, encoding a member of the peptide-release-factors protein family were found in patients with a combination of LS, OA, and PEO [124].

### 2.6. Complex V Defects (MTATP6)

ATP synthase (CV) comprises an integral membrane component F0 and a peripheral moiety F1. Only two F0 proteins (ATP6 and 8) are encoded by mtDNA [125].

Different heteroplasmic mutations in *MTATP6* (9176T → C, 9185T → C, 9176T → G) have been associated with LS [126]. Depending on the percentage of heteroplasmy, the most frequent mutation, a 8993T>G transversion, can lead to severe, early-onset maternally inherited LS (MILS), or to milder, juvenile or adult-onset NARP (neurogenic muscle weakness, ataxia, retinitis pigmentosa; OMIM 551500). MILS typically occurs when the m.8993T>G mutation is >80–90%, whereas NARP is commonly associated with percentages around 50–60%. A spectrum of conditions of progressively increasing severity can occur with intermediate percentages of heteroplasmy. The second most frequent mutation is a transition T → C in the same position, associated with juvenile Leigh or NARP syndromes [127]. The percentage of heteroplasmy in both mutations is similar in different tissues, including chorionic villi; this observation has prompted several centers, including ours, to perform genetic prenatal diagnosis in pregnant women carrying the NARP mutations, with a high degree of predictive reliability. See Table 5.

## 3. Mutations in mtDNA Maintenance Genes

### 3.1. Alpers–Huttenlocher Hepatoencephalopathy

#### Clinical Presentation

Alpers–Huttenlocher syndrome (AHS) has recently been recognized as a mitochondrial disease, but was described in the early 1930s by Dr Alpers as a severe diffuse of progressive poliodystrophy in the cortex and deep cerebral nuclei characterized by spongiotic necrosis of the gray matter. Huttenlocher associated this neuropathological entity with liver involvement, ranging from increased levels of plasmatic hepatic enzymes to severe liver failure. The onset is usually in infancy or early childhood, and sometimes in adolescence [128,129]. The initial clinical features are characterized by severe hypotonia and refractory seizures [130]. Status epilepticus is a frequent, often fatal, complication; however, valproate should be avoided as it may trigger hepatic failure [131]. In most patients, the course of the disease is rapidly progressive, the exitus occurring usually before three years of age. Brain MRI is hallmarked by severe, progressive cortical and subcortical atrophy, also with involvement of deep gray structures, e.g., the thalami (Figure 4).

Neuropathology reveals severe supra- and infratentorial poliodystrophy with multiple lesions affecting both cortical and subcortical grey structures and adjacent white matter (Figure 5). Microscopic examination reveals the presence of focal areas of spongiotic degeneration with apoptosis, widespread, patchy involvement of the cortex with astrocytosis, vacuolization, neuronal loss, and capillary proliferation [132,133].

AHS patients have the combination of neurological lesions and symptoms with progressive cirrhosis of the liver leading to liver failure, which is typical of this disorder. The onset is frequently very early in infancy, following a disease-free interval of a few months after birth. The neurological presentation is dominated by intractable multiple focal epilepsy frequently leading to *epilepsia partialis continua*. However, a constellation of additional neurological symptoms can ensue during the course of the disease, before the exitus in a vegetative state, including psychomotor regression, spastic tetraparesis, and cerebellar ataxia. Profound depletion of mtDNA in the liver, rather than accumulation of multiple deletions, has been reported in a few cases, while mtDNA content is normal in the muscle. Depletion of mtDNA (see below) has been hypothesized, but not firmly established, in the brain of *POLG*-positive AHS patients, while it has been documented in the liver.

### 3.2. Molecular Findings

AHS is an autosomal recessive disease associated with MDS in the liver and sometimes muscle. In liver mtDNA, the amount can be as low as <10% of the normal. Mutations in *POLG* determining profoundly reduced activity of mtDNA polymerase γ are the most frequent cause of AHS. More than a hundred mutations have been identified in POLG, associated with a number of different clinical presentations. AHS is at the end of a clinical spectrum that also includes juvenile onset spinocerebellar ataxia and epilepsy (SCAE) syndrome, its variant sensory-ataxic neuropathy, deafness and ophthalmoplegia (SANDO, OMIM 607459), and adult-onset, autosomal recessive (ar) or dominant (ad) progressive external ophthalmoplegia (PEO, OMIM 157640, and 258450), with or without additional features, such as generalized myopathy, peripheral sensory–motor neuropathy, parkinsonism, bipolar affective disorder, and ovarian failure with precocious menopause. Whilst liver mtDNA depletion is the molecular hallmark of AHS, and has also been occasionally documented in SCAE, the *POLG*-associated PEO syndromes are characterized by the accumulation of multiple mtDNA-deleted species in skeletal muscle (in brain also). This is a common molecular signature of mendelian PEO, irrespective of the primary genetic cause.

Mutations in *POLG* are relatively specific to different clinical presentations. In particular, AHS is frequently, but not exclusively, associated with the presence of two mutations, either pAla467Thr or p.Trp748Ser [134]. In general, one allele carries either mutation, whereas the other contains mutations in other amino acid residues. In most cases, but not always, the second mutation is in the polymerase domain of the protein, but many exceptions have been reported. Neuropathological investigations of an ataxic patient who carried the p.Ala467Thr mutation of *POLG* revealed sensory involvement of both peripheral and central axons as well as neuronal loss of the sensory ganglia [135]. In Finland, carrier frequency for the p.Trp748Ser mutation was estimated to be 1:125 [136], while in Norway, it is higher (1:100) and the combined carrier frequency for p.Trp748Ser and p.Ala467Thr is 1:50.

Early-onset hepatoencephalopathy with MDS can be caused by two additional nuclear genes, besides *POLG*: deoxyguanosine kinase (*DGUOK*) [137] and *MPV17* [72]. In mutations of the latter two genes, hepatic MDS is the predominant molecular trait and the major clinical problem, whereas the neurological impairment occurs later. The hepatic involvement determines severe, neonatal or very early-onset metabolic acidosis, severe hypoglycemic episodes, and eventually liver failure and cirrhosis, usually causing early fatal outcome. *DGUOK* encodes dGK, the mitochondrial deoxyguanosine kinase that is part of the nucleotide mitochondrial salvage pathway by phosphorylation of purine nucleosides. Impairment of dGK activity, and of its partner enzyme specific to pyrimidine nucleosides, thymidine kinase 2, TK2 [138], leads to severe shortage and imbalance of nucleotides, the ‘building blocks’ of mtDNA synthesis, which ultimately causes MDS. Whilst mutations in dGK are linked to hepatocerebral MDS (variant 3, OMIM 251880), mutations in *TK2* are associated with early-onset myopathy, or encephalomyopathy due to tissue-specific MDS (variant 2, OMIM 609560). Hypomorphic *DGUOK* and *TK2* alleles have been found in PEO encephalomyopathy with accumulation of multiple mtDNA deleted species (OMIM 617070 and 617069, respectively). Mutations in *MPV17*, encoding a protein of unknown function of the IMM, are responsible for a peculiar form of hepatocerebral MDS (variant 6, OMIM 256810), which also includes Navajo familial neurohepatopathy, NNH, a condition restricted to the Navajo population, caused by a ‘founder’ missense *MPV17* mutation, the c.Trp50Gln [73]. Less drastic *MPV17* mutations are responsible of axonal CMT2EE (OMIM 618400).

Another, exceptionally rare, early-onset encephalopathic variant of hepatoencephalopathic mtDNA depletion is infantile onset spinocerebellar ataxia (IOSCA or MTDPS7, OMIM 271245), a disease due to a single, recessive mutation (p.Tyr508Cys) in the *C10ORF2* gene, encoding the Twinkle helicase, which is part of the Finnish disease heritage [63]. These patients are characterized by a severe neurodegenerative disorder with a combination of ataxia, athetosis, hypotonia, sensorineural deafness, and severe epilepsy. They develop progressive atrophy of the cerebellum, brainstem, and spinal cord, and a sensory axonal neuropathy, associated with mtDNA depletion in the brain and liver [136]. Neurons of the cerebellum and frontal cortex have decreased activity of CI. A different homozygous mutation (T457I) in *C10ORF2* has also been identified in three Algerian consanguineous patients with a similar condition, including severe hepatocerebral phenotype characterized by neonatal hypotonia, mild liver insufficiency, increased serum and cerebrospinal fluid (CSF) lactate, psychomotor regression, seizures, and peripheral neuropathy [65].

An additional entity associated with mtDNA depletion is due to mutations in *RRM2B*, encoding the ribonucleotide reductase regulatory TP53 inducible subunit M2B [139]. RRM2B is part of a heterotetrameric enzyme, which catalyzes the conversion of ribonucleoside diphosphates to deoxyribonucleoside diphosphates. The product of this reaction is necessary for DNA synthesis. Mutations in *RRM2B*, have been associated with MDS, but also with PEO-5, and a variant of MNGIE. The MDS is usually severe, the recessive syndrome is neonatal or early-onset, and the most affected tissues are the skeletal muscle, the CNS, frequently with intractable epilepsy, OA, feeding difficulties, failure to thrive, hypotonia, lactic acidosis, and massive aminoaciduria consistent with severe proximal tubulopathy.

Finally, DNA ligase III (LIG3) is essential for mitochondrial DNA integrity but dispensable for nuclear DNA repair [140]. Inactivation of LIG3 in mouse CNS has been shown to result in mtDNA loss leading to profound mitochondrial dysfunction, disruption of cellular homeostasis, and incapacitating ataxia. Recently, recessive mutations in LIG3 have been associated with a MNGIE-like syndrome with mild mtDNA depletion [141], but also with a fulminant neonatal multisystem MDS, with predominantly neurological failure [141]. The human LIG3 gene contains two putative starting codons; the upstream ATG is the translation initiation site for the mitochondrial isoform. The DNA sequence between the two ATGs encodes an amphipathic helix, which resembles already known MTS peptides. Interestingly, the severe MDS form was associated with compound heterozygosity, including a p.Trp29* mutant allele, located in the MTS; hence, in this allele, the creation and translation of a transcript may still start from the second ATG, thus allowing the synthesis of the nuclear isoform, despite the presence of an early stop codon, which impairs only the mitochondrial isoform.

### 3.3. Leukoencephalopathy

#### Clinical Presentation

Generalized white matter degeneration has been observed in an increasing number of patients with mitochondrial encephalopathy. In a series of over 300 pediatric cases, leukoencephalopathy was the predominant or exclusive MRI feature in approximately 20% of the patients with severe, relatively isolated white matter degeneration, and the virtual absence of any significant alteration in deep brain nuclei or brainstem. In some children, large cystic lesions were observed within the white matter, whereas other cases were characterized by progressive, albeit late, vacuolization. In still other cases, typically associated with PDHC deficiency, central hypomyelination was concomitant to cortical developmental abnormalities, e.g., micropolygyria.

Mitochondrial infantile leukoencephalopathy is often associated with defects of complex I or complex II, but occasionally also with COX deficiency. For instance, mutations in *SURF1* have been occasionally detected in predominantly leukoencephalic lesions. Irrespective of the biochemical defect, two major clinical presentations have been observed: (i) infants with very early psychomotor delay, failure to thrive, and growth impairment, suffering a rapid downhill course resulting in severe spastic quadriparesis and cognitive impairment; (ii) children characterized by a disease-free period during the first years of life, followed by acute onset of focal motor disturbances, seizures, and a slowly progressive downhill course, with impaired motor abilities, but relative preservation of cognitive functions.

The diagnosis is mainly based on the MRI pattern, and must differentiate mitochondrial cases from the ample spectrum of other early-onset leukodystrophies, including Alexander’s disease, Canavan’s disease, megalencephalic leukoencephalopathy with subcortical cysts, and vanishing white matter. Brain proton transfer mass resonance spectroscopy (H^+^-MRS) may be useful in differential diagnosis. In mitochondrial diseases (H^+^-MRS) can detect increased concentration of lactate in brain regions not yet morphologically altered, that can therefore fail to be detected by MRI. However, the presence of an H^+^-MRS lactate peak is not specific to mitochondrial related pathogenesis, as it can be detected also in the active phase of other inherited leukoencephalopathies, or in ischemic and inflammatory lesions. However, an H^+^-MRS peak corresponding to accumulated succinate is a hallmark of CII deficiency, particularly in *SDHAF1* mutant patients [142].

A rather specific mitochondrial white matter disease of late childhood or young adulthood is leukoencephalopathy with brainstem and spinal cord involvement and lactate elevation (LBSL) [143]. LBSL is caused by mutations in the gene encoding *DARS2*, the mitochondrial aspartyl-tRNA synthetase. The initial signs usually consist of gait disturbances, followed by slowly progressive cerebellar ataxia, pyramidal signs, and sensory abnormalities due to degeneration of the ascending dorsal tracts of the spinal cord. The diagnosis relies on a typical MRI sign characterized by signal abnormalities of the cerebral white matter, dorsal columns and corticospinal tracts, pyramids, cerebellar peduncles, intraparenchymal portion of the V cranial nerve, posterior arm of the internal capsule, and splenium of the corpus callosum.

### 3.4. Molecular Findings

MRC defects are relatively frequent and should be sought in patients with leukodystrophy. Leukoencephalopathy, caused by isolated CI deficiency, can be due to mutations in structural subunits or assembly factors of the complex. For instance, mutations in *NDUFV1*, encoding the 51 kDa subunit, or *NDUFS1*, encoding the FMN-associated 70 kDa subunit of complex I, can cause leukodystrophy and myoclonic epilepsy [20], in addition to LD; whereas a mutation in *NUBPL*, encoding a protein that incorporates the Fe–S clusters into CI subunits, has been found in a single patient with leukodystrophy and elevated lactate in the CSF [144].

Accumulation of lactate and succinate in leukodystrophic white matter is the H^+^-MRS hallmark of severe reduction of CII activity and amount, due to mutations in *SDHAF1*, encoding a specific CII assembly factor (Figure 3C,D) [142].

Leukodystrophic features have occasionally been reported in subjects with isolated COX deficiency. A loss-of-function mutation in *SURF1*, a gene usually associated with LD (see above), has been reported in isolated leukodystrophy, including degeneration of the corticospinal tracts [145]. Moreover, the only case so far reported to be associated with a mutation in the nuclear-encoded COX6B1 subunit [146] showed a combination of early-onset leukodystrophic encephalopathy, myopathy, and growth retardation with COX deficiency. However, a COX-related leukoencephalopathy was found in mutations of *APOPT1,* now known as *COA8* [147]. The product of this gene was originally considered an anti-apoptotic mitochondrial protein expressed in hypertrophic smooth muscle of atheromatous lesions in the mouse. However, recessive mutations in *COA8* were later found in two siblings affected by a peculiar cavitating supratentorial posterior leukodystrophy, which allowed the discovery of additional patients. The main clinical manifestations included spastic tetraparesis, ataxia, and sensorimotor polyneuropathy. The biochemical hallmark of *COA8* mutations is severe COX deficiency, usually triggered by intercurrent febrile episodes. The protein seems to be involved in a complex anti-stress mechanism, and it is in fact stabilized by increased ROS. In these conditions, the precursor *COA8* protein, that is usually eliminated by the ubiquitin–proteasome system, UPS, enters mitochondria, is cleaved and loses its mitochondrial targeting peptide, MTS, acting as a stabilizing and ‘protective’ factor specific to COX structural and functional integrity [148]. No obvious interactors have been found for *COA8,* but its absence is consistently associated with COX deficiency, and accumulation of the mitochondrial translation regulation assembly intermediate of cytochrome c oxidase (MITRAC), the first structure formed during the assembly of nascent COX. Interestingly, MITRAC can become an ROS producer after its metalation.

Leukodystrophic lesions have been reported rather frequently in a defect of mtDNA translation. For instance, severe infantile macrocystic leukodystrophy with micropolygyria and multiple MRC defects, was associated with a homozygous mutation in the gene encoding mitochondrial elongation factor Tu (EF-Tu) [87] with consequent impaired EF-Tu binding to its tRNA substrate.

As already mentioned, *DARS2* mutations can determine leukoencephalopathy with brainstem and spinal cord involvement and lactate elevation (LBSL) [149]. Almost all patients with LBSL are compound heterozygotes, sharing a complex rearrangement in one allele that involves a T–C stretch upstream from exon 3 (228-20/-21delTTinsC) and a second variable mutation. The 228-20/-21delTTinsC partially interferes with the splicing of exon 3, leading to frameshift and premature truncation (p.Arg76SfsX5) of a fraction of *DARS2* transcripts. The residual aliquot of normal *DARS2* transcript explains why LBLS is a slowly progressive condition, compared to early, sometimes fulminant syndromes due to mutations in other nuclear genes involved in mtDNA translation (e.g., *EFG1* and *EF-Tu*). Pathogenic mutations in each of the 19 nuclear genes coding for a mitochondrial aaRS have been reported [84,150,151,152,153,154]. Defects in the exclusively mitochondrial enzymes all have either homozygous or compound heterozygous presentations, giving rise to autosomal recessive disorders. Mutations in the dual-localized *GARS* and *KARS* genes were reported with both recessive and dominant inheritance, giving rise to different clinical presentations. Autosomal dominant mutations in *GARS* and *KARS* affect the PNS and are correlated with Charcot–Marie–Tooth disease type 2 (CMT2) [155]. Recessive mutations in these genes, however, produce phenotypes similar to those reported by mutations in exclusively mt-aaRSs [156,157]. Pathogenic mutations in human mt-aaRSs are listed in Table 6 as in [158]. Although genes for mitochondrial aaRSs are nucleus-encoded and ubiquitously expressed, mutations give rise to a variety of distinct phenotypes [84,150,151,152,153,154]. With a few exceptions detailed below, all mutations in a particular synthetase result in similar disease states. These effects are manifested mostly in the CNS but also in a variety of other tissues (Figure 6). The available data present a number of surprising contrasts that complicate simple hypotheses based on the linkage between defects in mitochondrial translation and a reduction in cellular ATP production. Tissue-specific developmental differences in energy requirements, connections with pathways for mitochondrial homeostasis associated with differences in intraorganellar localization, and alternative functions of the mitochondrial aaRS proteins are among the hypotheses currently under investigation.

### 3.5. Variability of Clinical Features

Disorders correlated with mutations in mitochondrial aminoacyl-tRNA synthetases (mt-aaRS) span a broad range, including diseases characterized by defined symptoms and/or neuroradiological features (e.g., LBSL), isolated clinical signs (e.g., non-syndromic hearing loss) to described syndromes (e.g., Perrault syndrome). Since the first description of a correlation between mutations in mt-aaRS–encoding gene and a human disease [149], the number of reported cases has increased steadily [158].

According to a recent interesting publication, which summarizes the complex nosological area associated with mutations in mt-aaRS [158], four main groups emerge (Table 6): mt-aaRSs with mutations leading to clinical manifestations (i) exclusively in the CNS; (ii) in the CNS and another system; (iii) in the CNS or another system, and (iv) a system other than the CNS. The main features of each category and genes are summarized in Table 6.

However, heterogeneity exists within these four groups (Figure 6). For example, among the mutations that affect the CNS, there is a strong correlation between early onset of disease and the severity of the clinical symptoms, illustrated by the contrast between *DARS2-*associated leukoencephalopathies, which present as LBSL disease and *RARS2*-associated epileptic encephalopathy, which presents as pontocerebellar hypoplasia type 6 (PCH6). LBSL patients usually develop movement problems during childhood or adolescence, but in some cases, the clinical manifestations do not appear until adulthood. Symptoms presented by individuals with LBSL are mainly spasticity (muscular stiffness) and ataxia (difficulty with coordinating movements). These conditions tend to affect the legs more than the arms. In the most severely affected patients, the use of wheelchair assistance is required [160]. In contrast, PCH6 patients manifest the symptoms soon after birth with, in most cases, intractable seizures and recurrent apnea [161]. Other neurological signs include generalized hypotonia, microcephaly (unusually small head size, caused by impaired growth of some parts of the brain), lethargy, poor suckling, and poor feeding. The most heavily affected patients live only into infancy or childhood, and they never achieve developmental milestones [162]. Patients with *RARS2* mutations usually manifest symptoms soon after birth, with severe seizures that tend to evolve into epileptic status. In contrast, the later the symptoms become present in LBSL patients, the milder the symptoms (e.g., weakness in the lower limbs).

This relationship between early onset and severity of symptoms is observed in other cases as well. In patients with *YARS2* mutations presenting MLASA, mortality was usually a consequence in patients with early onset. However, some exceptions have been noted; for instance, one *YARS2*-related patient with early onset showed spontaneous improved muscle strength and stamina at the age of 17 years and no longer required blood transfusions (which had previously been given every 6 weeks) [163].

Although this categorization is meant to point out distinct classes of mt-aaRS–related disease, it remains unclear whether the enzymes belonging to each of the groups described above have similar cellular properties that explain similarities in clinical phenotypes.

### 3.6. Encephalo–Cardiomyopathy

#### Clinical Presentation

Encephalo–cardiomyopathy is a severe, usually fatal, mitochondrial syndrome of early infancy. Children are frequently critically ill at birth, being affected by severe heart failure with lactic acidosis. They usually show hypertrophic cardiomyopathy, severe myopathy and/or central hypotonia, failure to thrive, and respiratory distress. Other signs may be present, such as microcephaly, hepatomegaly, facial dysmorphism, e.g., low-set ears, retrognathia, and prominent nasal bridge with hypertelorism. The MRI is rather unspecific but may show abnormal signal intensity in the periventricular white matter, and occasionally, lesions of the deep gray nuclei. The clinical course may be fulminant with fatal outcome in the neonatal period. The patients who survive the first months of life are characterized by psychomotor delay, with a variable set of other signs, including oculomotor disturbances, e.g., nystagmus, cognitive impairment, ataxia, and myopathy. High lactate levels may be detected in plasma, CSF, and urine. In a specific condition, X-linked Barth syndrome, severe dilating cardiomyopathy with myocardial non-compaction and fluctuating neutropenia is biochemically characterized by increased excretion of 3-methylglutaconic acid.

As with most of the early-onset mitochondrial disorders, the muscle biopsy is usually scarcely informative and only biochemical investigation of the MRC in muscle or fibroblasts can lead to the diagnosis.

### 3.7. Molecular Findings

Isolated defects of CI, CIV, or CV are the most frequent biochemical abnormalities. However, a homozygous mutation in *SDHA*, encoding the largest subunit of CII (G555E), was previously associated with LS, but has later been found in two large consanguineous families with neonatal isolated cardiomyopathy [164].

CI deficiency has been reported in patients with encephalo–cardiomyopathy, carrying mutations in genes encoding either structural CI subunits (*NDUFS2*, *NDUFV2*, and *NDUFA11*) or specific CI assembly factors (*NDUFAF4* and *ACAD9*). Single cases/families have been reported for *NDUFV2, NDUFA11*, and *NDUFAF4*, whereas several mutations have been found in *NDUFS2* [165] and *ACAD9* [166]. *ACAD9* encodes a poorly defined component of the mitochondrial acyl-CoA dehydrogenase family, possibly involved in beta oxidation of fatty acids; therefore, ACAD9 seems to have a double function in the management of fatty acids as well as in the interaction with several CI assembly factors. *ACAD9* mutations are in fact associated with severe, neonatal, sometimes fatal, lactic acidosis, followed by hypertrophic cardiomyopathy. Although cardiac failure may be the predominant symptom in the surviving patients, encephalopathy with mental retardation and poor growth have also been reported [167].

*SCO2,* in concert with *SCO1*, encodes two proteins enabling the first two subunits of complex IV to be incorporated into the holoprotein. Eight mutations in *SCO2* have been described in patients with fatal infantile encephalo–cardiomyopathy and COX deficiency [168]. Interestingly, all patients reported were compound heterozygotes; even more remarkably, one particular mutation, p.Glu140Lys, was present in all affected individuals. Mutations in another assembly factor for CIV, *COX15,* can also cause, albeit less consistently, fatal infantile hypertrophic cardiomyopathy.

Mutations in *TMEM70,* encoding a putative assembly factor of CV, were found in patients, mostly of Gipsy origin, with cardiomyopathy and isolated deficiency of ATP synthase [169]; the prevalent homozygous mutation, a A → G transition in intron 2 of the *TMEM70* gene, results in aberrant splicing and loss of the mRNA transcript. This mutation is associated with a high degree of intrafamilial variability in the severity of symptoms.

A deficiency of ATP synthetase was also reported in two siblings with lactic acidosis, hypertrophic cardiomyopathy, and muscular hypotonia [170]; this was due to a homozygous mutation in the *SLC25A3* gene, the mitochondrial phosphate carrier. The mutation affects the alternatively spliced exon 3A, expressed in muscle.

Other mitochondrial disorders with cardiac involvement, but without a specific biochemical deficiency, include a mutation of *DNAJC19*, which encodes a putative mitochondrial import protein. The mutation causes dilated cardiomyopathy with ataxia [171]. In addition to mutations of carrier or import proteins, alteration of the lipid milieu of the IMM, which is a unique structure for its exclusive content of cardiolipin, can also determine OXPHOS dysfunction. For instance, Barth syndrome (OMIM 302060) is due to mutations in *TAZ* (or *G4*.*5*), an X-linked gene encoding a cardiolipin-specific acyl–coenzyme A synthetase (tafazzin) involved in the biosynthesis and structural maturation of this crucial phospholipid of the IMM. Accordingly, cardiolipin is markedly decreased in skeletal and cardiac muscle and in platelets from affected patients [172].

### 3.8. Other Disorders

Fulminant hepatocerebral failure has been reported in consanguineous patients with neonatal ketoacidotic coma and profound COX deficiency. Two allelic mutations, a 2 bp frameshift deletion and a P174L, were identified in the protein encoded by *SCO1* [119] (standing for synthesis of cytochrome oxidase 1), a COX assembly factor that promotes the incorporation of copper atoms in the catalytic subunits COX1 and COX2 of nascent complex IV.

An intriguing syndrome has been associated with a previously unknown, putative E3 ligase, FBXL4, localized in the outer mitochondrial membrane (OMM). Autosomal recessive *FBXL4* mutations are associated with early-onset lactic acidemia, hypotonia, and developmental delay caused by severe encephalomyopathy consistently associated with progressive cerebral atrophy and variable involvement of the white matter, deep gray nuclei, and brainstem structures. A wide range of other multisystem features can variably be seen, including dysmorphism, skeletal abnormalities, poor growth, gastrointestinal dysmotility, renal tubular acidosis, seizures, and episodic metabolic failure. Mitochondrial respiratory chain deficiency is present in muscle or fibroblasts, together with a markedly reduced oxygen consumption rate and hyper-fragmentation of the mitochondrial network in cultured cells. In muscle and fibroblasts from several subjects, a substantially decreased mtDNA content has been observed. FBXL4 is a member of the F-box family of proteins, some of which are involved in phosphorylation-dependent ubiquitination and/or G protein receptor coupling. A *FBXL4* knockout mouse shows reduced mitochondrial mass, suggesting a role for the protein in controlling organellar biogenesis possibly by acting on mitochondrial autophagy [173]. Likewise, mutations in another putative E3 ligase of the OMM, Fbxo7/PARK15, have better-defined roles acting as part of a Skp1-Cul1-F box protein (SCF)-type E3 ubiquitin ligase as well as having SCF-independent activities. Mutations within *FBXO7* have been found to cause an early-onset Parkinson’s disease, and these are found within or near to its functional domains, including its F-box domain (FBD), its proline rich region (PRR), and its ubiquitin-like domain (Ubl) [174].

The systematic exploitation of deep sequencing technologies has allowed neuroscientists to identify de novo heterozygous mutations associated with severe, early-onset neurodegeneration. This is the case of *de novo SLC25A4* mutations affecting the gene encoding the ANT1 mitochondrial translocator in seven unrelated infants [175]. All affected individuals presented at birth, were ventilator dependent and, where tested, revealed severe combined mitochondrial respiratory chain deficiencies associated with a marked loss of mtDNA copy number in skeletal muscle. Strikingly, an identical c.239G>A (p.Arg80His) mutation was present in four of the seven subjects, and the other three case subjects harbored the same c.703C>G (p.Arg235Gly) mutation. Analysis of skeletal muscle revealed a marked decrease of AAC1 protein levels and loss of respiratory chain complexes containing mitochondrial DNA-encoded subunits. This is a fatal condition, variably combining neurological signs (hypotonia, hyporeflexia, and floppiness), with respiratory muscle insufficiency, and in some subjects, hypertrophic cardiomyopathy. Likewise, de novo heterozygous mutations can affect *DNM1L*, encoding a protein with a major role in mitochondrial fission, usually associated with severe, infantile encephalopathy, whereas transmissible, recessive *DNM1L* mutations cause a form of optic atrophy (OA5) (OMIM *603850) [176]. Incidentally, an expanding number of mutations involve other genes associated with mitochondrial dynamics, such as *MFN2*, encoding mitofusin 2, promoting fusion of the mitochondrial outer membrane, heterozygous mutations of which are responsible for CMT2B (OMIM 608507) or *OPA1,* a gene responsible for autosomal dominant optic atrophy type 1 (ADOA1), but also of a syndromic form of dominant optic atrophy with encephalopathic PEO and multiple mtDNA deletions (OMIM *605290) [176].

Finally, two male infant patients who were given a diagnosis of progressive mitochondrial encephalomyopathy on the basis of clinical, biochemical, and morphological features [177]. These patients were born from monozygotic twin sisters and unrelated fathers, suggesting an X-linked trait. Fibroblasts from both showed reduction of respiratory chain (RC) CIII and CIV, but not of CI activities. These laboratory features were associated with accumulation of multiple, asymmetric lesions in the striatum, temporal cortex, and insula, and other brain districts, with parallel neurological failure. A disease-segregating mutation was found in the X-linked *AIFM1* gene (Xp26.1), encoding the apoptosis-inducing factor (AIF) mitochondrion-associated 1 precursor (OMIM *300169) that deletes arginine 201 (R201 del) [177]. Under normal conditions, mature AIF is an FAD-dependent NADH oxidase of unknown function and is protruding from the IMM to the mitochondrial intermembrane space (this form is called AIF(mit). Upon apoptogenic stimuli, a soluble form (AIF(sol)) is released by proteolytic cleavage and migrates to the nucleus, where it induces ‘parthanatos,’ i.e., caspase-independent fragmentation of chromosomal DNA. In vitro, the AIF(p.Arg201del) mutation decreases stability of both AIF(mit) and AIF(sol) and increases the AIF(sol) DNA binding affinity, a prerequisite for nuclear apoptosis. Additional male infants with *AIFM1* mutations have subsequently been identified by different groups. Interestingly, less drastic mutations in *AIFM1* were also found to be responsible of Cowchock syndrome (OMIM 310490), also known as X-linked recessive Charcot–Marie–Tooth disease-4, with or without cerebellar ataxia (CMTX4) [178]. Besides playing a key role in the execution of caspase-independent cell death, AIF has emerged as a protein critical for cell survival. Analysis of in vivo phenotypes associated with AIF deficiency and defects, and identification of its mitochondrial, cytoplasmic, and nuclear partners revealed the complexity and multilevel regulation of AIF-mediated signal transduction and suggested an important role of AIF in the maintenance of mitochondrial morphology and energy metabolism. The redox activity of AIF is essential for optimal oxidative phosphorylation [179,180].

These are some examples of the significant number of early-onset mitochondrial encephalopathies that do not meet the definition of specific phenotypes, and must therefore be included in a miscellaneous group of diseases. Some are very rare, having been described only in single families or isolated populations. In spite of their rarity, these disorders illustrate well the complex relationship between mitochondrial dysfunction and human diseases.

### 3.9. Neurodegeneration Associated with Mitochondrial Impairment in Adult Patients

In general, organs with the highest demand for aerobic energy, such as skeletal muscle, brain, and heart, appear to be the most commonly affected by MRC dysfunction, although any, and indeed all, tissues can be involved. Post-mitotic cells, such as muscle cells, neurons, and pancreatic beta-cells, cannot eliminate energy-spent mitochondria through mitotic segregation and are therefore those most at risk from energy impairment. Mitochondrial disease can affect one tissue alone (e.g., pure myopathy, encephalopathy, or cardiomyopathy) or, more usually, a combination of tissues. In the pre-genetic era, classification of mitochondrial disorders was based on clinical, and later also morphological or biochemical results (especially in skeletal muscle biopsy), but are now conclusively established by genetic evidence. Similar to pediatric syndromes, mitochondrial conditions in adults are also grouped into a) pathogenic primary mtDNA mutations and b) pathogenic mutations of OXPHOS-related nuclear genes, which also include mutations in genes encoding factors deputed to replication, maintenance, and expression of mtDNA; in particular, mutations in components of the replicative apparatus can determine secondary mtDNA alterations, either qualitative (mtDNA multiple deletions) or quantitative (mtDNA deletion syndromes, MDS).

Although frequent, ataxia is rarely the only sign of mitochondrial impairment. Much more frequently, ataxia ensues in combination with other neurological and extra-neurological conditions, and indeed this unusual combination of multiorgan, or multi-neurological failure can orientate the clinician toward a mitochondrial disease. Mitochondrial ataxia may be cerebellar, sensory, or mixed (spinocerebellar). Central ataxia presents with nystagmus, dysarthria, and truncal unsteadiness, while sensory ataxia usually include gait in-coordination with frequent falls or need of an aid (e.g., a stick), worsening when patients close their eyes or when ocular fixation is compromised. Intermittent ataxia has been described in PDcE1α deficiency [181], and also in a patient with *MTATP6* gene mutations leading to NARP [182]. The presence of profound hypotonia, for example, associated with infantile mitochondrial encephalopathy, can mask, at least in part, the presence of ataxia.

Mitochondrial disease must be distinguished from other phenocopy disorders with similar features. The combination of ataxia and deafness (May–White syndrome), retinopathy and deafness (Usher syndrome), and myoclonic epilepsy and ataxia (Ramsay Hunt syndrome) are conditions where mitochondrial disease is one, but not the only, possible etiology. The differential diagnosis can be difficult, but a mitochondrial etiology may be suggested by the presence of associated signs, e.g., ophthalmoplegia (very common in mitochondrial disease), biochemical markers, e.g., high lactic acid levels in plasma and particularly in the CSF, or by the concomitancy of clinical involvement in extra-CNS systems, such as receptor deafness or diabetes mellitus.

### 3.10. Primary mtDNA Mutations

Of the many mtDNA-related syndromes reported over the last 30 years, four are relatively frequent and well characterized, both clinically and genetically: Kearns–Sayre syndrome (KSS); mitochondrial encephalopathy, lactic acidosis, and stroke-like episodes (MELAS); myoclonus epilepsy with ragged-red fibers (MERFF); and neurogenic weakness, ataxia, and retinitis pigmentosa (NARP). A growing number of mutations in mtDNA are being found; however, many are associated with encephalomyopathy. Unfortunately, correlation between phenotype and genotype is inconsistent such that one mtDNA mutation can cause several different phenotypes and different neurological manifestations, predominantly ataxia, can occur with several different mutations.

### 3.11. Kearns–Sayre Syndrome (KSS)

KSS is a severe, usually sporadic disorder characterized by the invariant triad of progressive external ophthalmoplegia (PEO), pigmentary retinopathy, and onset before age 20 [183]. Additional symptoms are poor growth, heart conduction blocks, increased CSF protein content, and a progressive cerebellar ataxia. Chronic progressive external ophthalmoplegia (PEO), an adult-onset disease with bilateral ptosis, ophthalmoplegia, and proximal myopathy, must be differentiated by juvenile or adolescent Kearns–Sayre syndrome (KSS), which is usually associated also with retinal dystrophy; in both syndromes, but particularly in KSS, the CNS is involved, typically with cerebellar ataxia, and, particularly in KSS, cognitive decline and movement disorders. In adult-onset PEO peripheral neuropathy can also occur.

KSS is characterized by distinctive neuroradiological abnormalities in the cerebellum and brainstem, but also in more rostral structures, for instance, the diencephalon (thalami), the striatum, and the supratentorial, especially subcortical, white matter [184] (Figure 7). In the brainstem, the mesencephalon is diffusely involved, including the red nuclei. In the cerebellum, the dentate nuclei and the dentatorubral fibers emerging through the superior cerebellar peduncle are the most heavily involved structures. Histopathology examination reveals neuronal degeneration and gliosis of the striatum and white matter spongiotic changes [185,186]. Loss of Purkinje cells is frequent in KSS, and severely reduced expression of mtDNA-encoded proteins can be detected in neurons of the dentate nucleus [187].

KSS and PEO are associated with large-scale rearrangements of mtDNA, usually deletions that are easily detected by Southern blot analysis of muscle mtDNA [183,188] and, more recently, deep sequencing technologies, which include both qualitative and quantitative approaches. In about 20% of cases, the disorder is due to a tandem duplication in which a deleted mtDNA species is joined to a wild-type species, or to a combination of both isolated deletions and tandem duplications [189]. With the exception of a few cases, rearrangements are sporadic traits, but in a multicenter study, a 4% recurrence risk was reported among the offspring of affected women [190].

### 3.12. Myoclonic Epilepsy and Ragged-Red Fibers (MERRF)

MERRF a maternally transmitted complex neurological entity is characterized by myoclonus, epilepsy, muscle weakness, deafness, progressive dementia, and cerebellar ataxia [191,192,193,194]. Ataxia may be the most relevant feature prominent feature of this disorder [195] at least initially, being usually cerebellar more than sensory, although spinocerebellar degeneration has also been described [196]. In late stages, the MRI displays diffuse cerebral and cerebellar atrophy, sometimes with calcification of the striatum, signal abnormalities in the dentate nuclei, alterations of the superior cerebellar peduncles, and inferior olives [184,191,197,198,199].

The commonest mtDNA mutation associated with MERRF is an A-to-G transition at position 8344 in the *MT-tRNA-K* gene [200]. A second mutation has been reported in the same gene, at position 8356 [201,202]. Correlation between disease severity, age of onset, mtDNA heteroplasmy, and reduced activity of MRC CI and CIV have been demonstrated. Even though genotype–phenotype correlation is tighter in this condition than in other mtDNA disorders, the m.8344A>G transition has also been reported in phenotypes as different as LS, isolated myoclonus, familial lipomatosis, and isolated myopathy [201,203]. MERRF ought to be differentiated from progressive myoclonus epilepsies, including Ramsay Hunt syndrome and Unverricht–Lundborg disease, in which cerebellar signs are prominent but neither mtDNA mutations nor RRF are detected [204,205].

### 3.13. Mitochondrial Encephalopathy, Lactic Acidosis, and Stroke-like Episodes (MELAS)

MELAS is typically defined by the presence of stroke-like episodes, due to focal brain lesions often localized to the occipital and parietal lobes, especially in the subcortical and cortical areas, and high lactic levels in plasms and CSF [206]. Other CNS signs, usually later in the course of the disease, include dementia, whereas recurrent headache and vomiting, focal or generalized seizures, and deafness may be initial, even prodromal symptoms. Ataxia has been observed in some patients [207].

Brain MRI (Figure 8) typically shows the failure of the abnormalities in the cerebellum, and supratentorial structures, to correspond to well-defined vascular territories, suggesting a primary parenchymal origin of the lesions, rather than ischemic abnormalities due to vascular accidents. Interestingly, cerebellar lesions can be detected by MRI well before the occurrence of stroke episodes, usually characterized by increased signal intensity in T2-weighted images of the cerebellar hemispheres [208].

Post-mortem studies have shown widespread infarct-like lesions associated with diffuse fibrillary gliosis in the cerebral and cerebellar white matter [209]. Histochemical and ultrastructural studies show accumulations of abnormal mitochondria in smooth muscle cells and the endothelium of cerebral and cerebellar blood vessels, suggesting a ‘mitochondrial angiopathy’. However, the presence of diffuse white matter gliosis of the CNS and cerebellar cortical degeneration, particularly in the granular cell layer, suggests morphologically widespread cellular dysfunction, not restricted to either neuronal or vascular derangement [210].

MELAS was first associated with a heteroplasmic point mutation, an A-to-G transition at position 3243 in the *MT-tRNA-L(UUR)* [206]. Other point mutations causing a MELAS-like disorder have also been reported (see www.mitomap.org) (available on 1 December 2021) although the m.3243A>G remains by far the most frequent.

### 3.14. Neurogenic Weakness, Ataxia, and Retinitis Pigmentosa (NARP)

NARP is a maternally inherited syndrome in which ataxia is a cardinal manifestation together with retinitis pigmentosa and a predominantly sensory neuropathy. Ataxia in NARP is considered mainly sensory, but an additional cerebellar component is also frequent and revealed by clinical and MRI findings, including moderate, diffuse cerebral and cerebellar atrophy and, in the most severely affected patients, symmetric lesions of the striatum [211].

NARP is most often associated with a heteroplasmic T-to-G transversion at position 8993 in the *MT-**ATPase6* subunit gene [212]. A transition affecting the same position (m.8993T>C) can also cause the same, albeit usually milder, clinical presentation [213]. Ragged-red fibers are consistently absent in the muscle biopsy. Disease severity correlates with the degree of heteroplasmy with higher mutant load leading to the more severe and early-onset mitochondrial Leigh phenotype (MILS) [214] (see also below). MILS (and NARP as well) have been reported with a spectrum of *MT-ATPase6* mutations over recent years, also described in association with other mutations of the *MT-ATPase6* gene, e.g., 9176T-> C [215,216]. NARP and MILS may coexist in the same family, depending on the heteroplasmic mutation load [217].

### 3.15. Other mtDNA Mutations

More than 100 mutations involving the mitochondrial genome are now identified (www.mitomap.org, accessed on 20 December 2021). Many mutations cause ataxia, some as a major manifestation, but most as part of a multisystemic disorder. The majority of point mutations are reported in one or a small number of families and, as stated earlier, no consistent genotype/phenotype correlation is seen, especially with mutations involving the mitochondrial tRNA. The presence of additional clinical features, such as deafness, epilepsy (particularly myoclonic epilepsy), myopathy, and PEO is common and should prompt the physician to consider mitochondrial disease as part of the differential diagnosis. The four examples given above encapsulate the type of disorder seen with mtDNA defects.

### 3.16. Defects of Nuclear Genes

Mitochondrial DNA encodes just 13 proteins, and all of these are components of the MRC complexes. All other proteins, whether they are components of the respiratory chain involved in its biogenesis or in mtDNA homeostasis, intramitochondrial protein translation, or another mechanism, must therefore be encoded by nuclear genes. Defects involving these gene products can be divided into six broad categories (Table 1): genes encoding factors affecting mitochondrial DNA maintenance; protein-encoding OXPHOS subunit genes; genes encoding OXPHOS assembly factors; genes encoding biosynthetic enzymes for lipids and cofactor genes encoding mtDNA translation factors; factors involved on mitochondrial detoxification, and, lastly, those indirectly related to mitochondrial oxidative phosphorylation. We have already discussed most of these categories in previous chapters of this review, including the nuclear genes controlling the integrity of the mitochondrial genome [218].

### 3.17. TYMP, Encoding Thymidine Phosphorylase (*TP*)

*TYMP* mutations are a paradigmatic genetically determined example of toxic impairment of mitochondrial metabolism, particularly mtDNA integrity. Defects in *TYMP*, encoding thymidine phosphorylase (TP) are rare and cause a specific syndrome, mitochondrial neurogastrointestinal encephalomyopathy (MNGIE) [219]. MNGIE also includes ophthalmoparesis, leukoencephalopathy, peripheral neuropathy with sensory ataxia [74,220], in addition to prominent gastrointestinal symptoms, including painful intestinal dysmotility leading to cachexia. Imaging studies show marked leukoencephalopathy, due to disruption of the blood–brain barrier, but the brain structures are otherwise intact.

Thymidine phosphorylase, an extra-mitochondrial enzyme, absent in skeletal muscle, is key in the catabolism of thymidine, and loss of enzyme activity leads to accumulation of pyrimidine nucleosides (deoxyuridine and deoxythymidine) but reduced amount of the other pyrimidine, deoxycytidine. This pyrimidine nucleotide imbalance is believed to interfere with the overall nucleotide pools, particularly that present within mitochondria, which are more dependent for DNA synthesis on the nucleoside salvage pathways than de novo synthesis, as occurs predominantly for the synthesis of nucleic acids in the cell nucleus [221]. Pyrimidine imbalance is mutagenic on mtDNA, with molecular instability that can manifest either as a quantitative loss (depletion) or a qualitative disturbance, multiple mtDNA deletions [222] as well as accumulation of somatic point mutations. As already mentioned, MNGIE phenocopies have been associated with mutations in recessive mutations in *LIG3*, as well as *RRM2B,* and *POLG* (MZ, personal observation).

### 3.18. C10Orf2, Encoding Twinkle, the mtDNA Helicase

Mutations in *C10Orf2* the gene encoding the mitochondrial helicase, Twinkle, are associated with adult-onset PEO inherited as an autosomal dominant trait. Typically, multiple mtDNA deleted species are detected in skeletal muscle. Sensory or spinocerebellar ataxia have been reported in adults and children, respectively [223]. Ataxia [224] may determine gait unsteadiness associated with a predominantly sensory neuropathy and MRI can show only supratentorial cortical atrophy.

As already mentioned, autosomal recessive infantile-onset spinocerebellar ataxia (IOSCA; OMIM 271245) mainly comes from Finnish children that usually present between 1 and 2 years of age with a progressive ataxia. Athetosis, muscular hypotonia, ophthalmoplegia, deafness, and sensory neuropathy develop, and epilepsy is a late manifestation. Brain MRI shows cerebellar and brainstem atrophy. Postmortem investigations show lesions similar to those seen in Friedreich ataxia [225]. A homozygous founder mutation determining a Y508C replacement in Twinkle was identified in all cases except one in which the Y508C was associated with a silent coding region cytosine-to-thymine transition. This allele was expressed at reduced levels, meaning that the individual was dependent on the Y508C allele.

### 3.19. POLG Encoding the Catalytic Subunit of mtDNA Polymerase, (POLγA)

As described above, disorders arising from mutations in the *POLG* gene are numerous and varied. As for Twinkle and ANT1, *POLG* mutations were first identified in patients with autosomally inherited PEO [226] and associated with the accumulation of multiple mtDNA deletions in skeletal muscle and other organs. Subsequently, other phenotypes were identified, and in several of these, ataxia is a major manifestation.

Adult-onset PEO associated with *POLG* mutations are frequently complicated by neurological symptoms, including spinocerebellar ataxia, peripheral motor neuropathy, extrapyramidal signs (parkinsonism), and psychiatric abnormality (unipolar or bipolar affective disorder) [227]. Recessive, dominant, and sporadic inheritance modalities have been reported. The combination of sensory ataxic neuropathy, dysphagia, and ophthalmoplegia (SANDO) was described in 1997 [228] and shown later to be due to *POLG* mutations [229].

A variant of SANDO, characterized by a spinocerebellar syndrome due to *POLG* mutations, has emerged as one of the commonest forms of recessively inherited ataxia, especially in Scandinavia and Finland [134,136,230]. This syndrome usually begins in the teens either with migraine-like headache and/or epilepsy or with a progressive ataxia, which combines sensory and cerebellar dysfunction. Myoclonus is also frequent and patients with epilepsy may frequently develop severe, sometimes fatal, *status epilepticus* or *epilepsia partialis continua*. Brain MRI (Figure 9) displays high signal lesions in T1-weighted sequences, involving the thalamus and central cerebellar white matter and occasionally the inferior olive. Cerebellar atrophy and infarct-like occipital lesions (laminar necrosis) are also frequent. Limitation of the eye movements can be seen early in the course of the disease, but complete ophthalmoplegia occurs as a late, end-stage sign. These spinocerebellar with ataxia and epilepsy (SCAE) patients display an exquisite sensitivity to valproate-induced hepatic toxicity, which may lead to irreversible hepatic failure.

Few post-mortem studies [231] show atrophy of the brainstem, cerebellum, and spinal cord, particularly in the dorsal columns. Patchy loss of Purkinje cells and severe degeneration of dentate nucleus and inferior olives were characterized by neuronal loss and gliosis.

Accumulation of multiple mtDNA deletions is not prominent in the skeletal muscle of SCAE patients, and may in fact be absent. No documentation is available on the presence of mtDNA abnormalities in the brain of these patients, although some reports indicate reduction of mtDNA copy number in the CNS.

### 3.20. Neurodegeneration Associated with Additional OXPHOS-Related Factors

Although defects involving these proteins do not necessarily lead to an OXPHOS defect, several cause ataxia. For example, frataxin, encoding by *FXN,* is a mitochondrial protein that is thought to play a role in intramitochondrial iron homeostasis linked to the biosynthesis of Fe–S redox clusters. Mutations in *FXN* [232] cause Friedreich ataxia (FA), one of the best known recessively inherited ataxias. FA is considered the commonest recessively inherited spinocerebellar ataxia, although in Finland and Norway, mutations in *POLG*, which can give a similar phenotype (IOSCA), are more common. It is unclear why the hypomorphic recessive alleles of *FXN* are specifically damaging the posterior root ganglia, the dorsal spinal columns and, albeit less prominently than other genetically determines spinocerebellar ataxia, the cerebellum and pons. However, FA should be considered a multisystem mitochondrial disorder, since the second, invariably affected, organ is the myocardium, heart failure being one of the most frequent causes of death in these patients, followed by the PNS, the endocrine pancreas (diabetes mellitus is a frequent finding in FA patients), and other organs. It is also important to emphasize that a number of other mitochondrial disorders, although much rarer than FA, are due to deleterious mutations in the complex enzymatic machinery leading to the synthesis of Fe–S clusters, including mutations in the ISCU complex, as well as in *ABC7*, an X-linked gene encoding the transporter of Fe–S clusters in the extramitochondrial space. *ABC7* mutations cause X-linked sideroblastic anemia and ataxia [233]. ABC7 belongs to the ATP-binding cassette transporters and is also involved in mitochondrial iron homeostasis. Many other intra-mitochondrial factors are involved in the completion of Fe–S clusters and in their incorporation in a wide spectrum of enzymes, including the mitochondrial aconitase (Aco2), as well as MRC CI, CII, and CIII. These factors can harbor mutations associated with disease in single enzymatic activities. In addition, Fe–S clusters are essential redox centers for numerous, diverse enzymatic activities in both the nucleus and the extra-mitochondrial cytoplasm, and therefore, defects of the Fe–S biosynthetic machinery upstream from and including the translocation of the clusters outside mitochondria (by the ABC7 transporter) can affect multiple cell pathways in both mitochondrial and extramitochondrial. For a more specific review of this important area of mitochondrial and cellular metabolism, the reader is referred to the work by Roland Lill and collaborators [234].

Defects in an increasing number of other nuclear genes are associated with a range of phenotypes, such as myopathy, encephalopathy, or liver dysfunction. The following examples demonstrate the range of features seen. Paraplegin, encoded by *SPG7*, is an ATP-binding metalloprotease involved in the ‘quality control’ of mitochondrial membrane-bound protein, and mutations in this gene cause AR hereditary spastic paraplegia [235]. X-linked deafness–dystonia syndrome results from the mutation of *DDP* [236], a transporter protein involved in the insertion of metabolite carriers into the inner mitochondrial membrane [237], or the product of *TIMM50*, one of the components of the TIM23 protein translocator of mitochondria [238]. Finally, mutations in *PITRM1*, the mitochondrial matrix protease, which digests the cleaved MTS and other oligopeptides within the inner mitochondrial compartment, have been associated with a complex neurological syndrome, invariably including mental retardation, psychosis, and ataxia [239,240]. Whilst the complete loss of *PITRM1* in a mouse model is embryonically lethal, the heterozygous individuals display neurodegeneration and accumulation of amyloid precursor protein (APP) and Aβ_40–42_ oligopeptides, suggesting a link between mitochondrial quality control and amyloidotic Alzheimer’s-like neurodegeneration, as also confirmed in brain human organoids [241].

## Figures and Tables

**Figure 1 cells-11-00637-f001:**
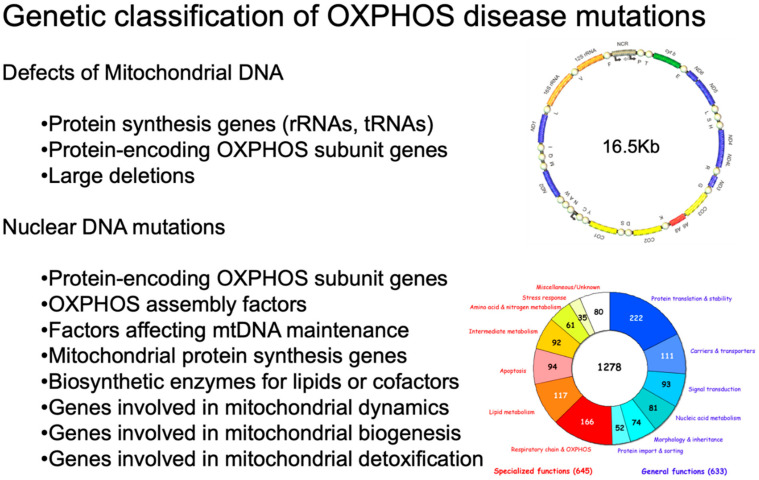
Genetic classification of OXPHOS disease mutations. On the upper right, a scheme of human mtDNA is depicted. The two ribosomal RNA genes are in dark yellow. CI genes are in blue; the CIII gene is in green; the CIV genes are in light yellow; and the CV genes are in red. The non-coding region (NCR) is in beige. The tRNA genes are represented by circles and designated according to the single-letter code of the corresponding amino acid. On the bottom right, a pie chart summarizes the current knowledge about the human mitochondrial proteome. Different colors indicate different categories of proteins. Adapted from [12] under the Creative Commons Attribution license.

**Figure 2 cells-11-00637-f002:**
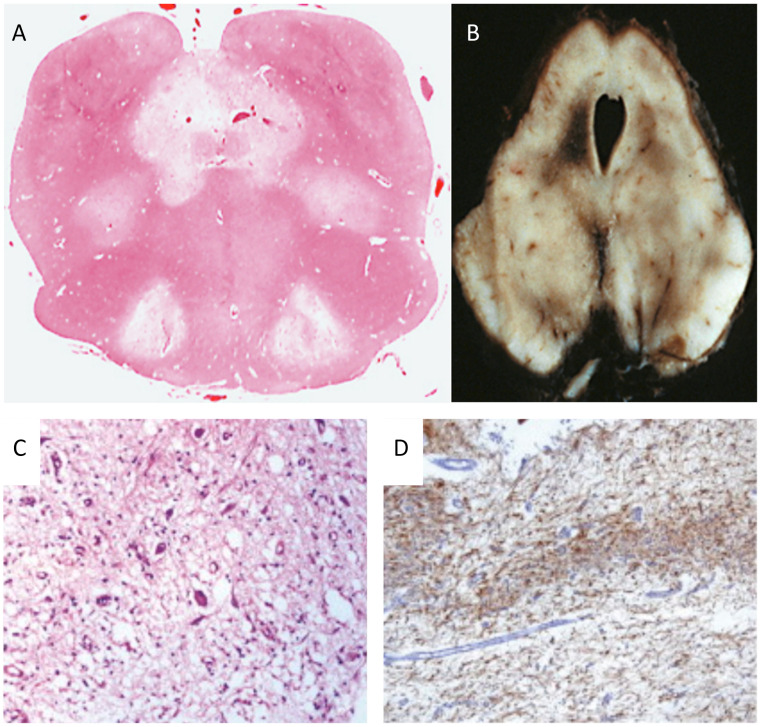
Neuropathology in Leigh disease. (**A**) necrotizing lesions in the medulla oblongata (H&E) and (**B**) mesencephalon (autoptic specimen). (**C**) H&E staining showing neuronal loss, microcystic cavitation of the neuropilum, vessel proliferation, and microgliosis; (**D**) GFAP immunohistochemistry shows marked gliosis in the dentate nucleus. Adapted and modified from: [114].

**Figure 3 cells-11-00637-f003:**
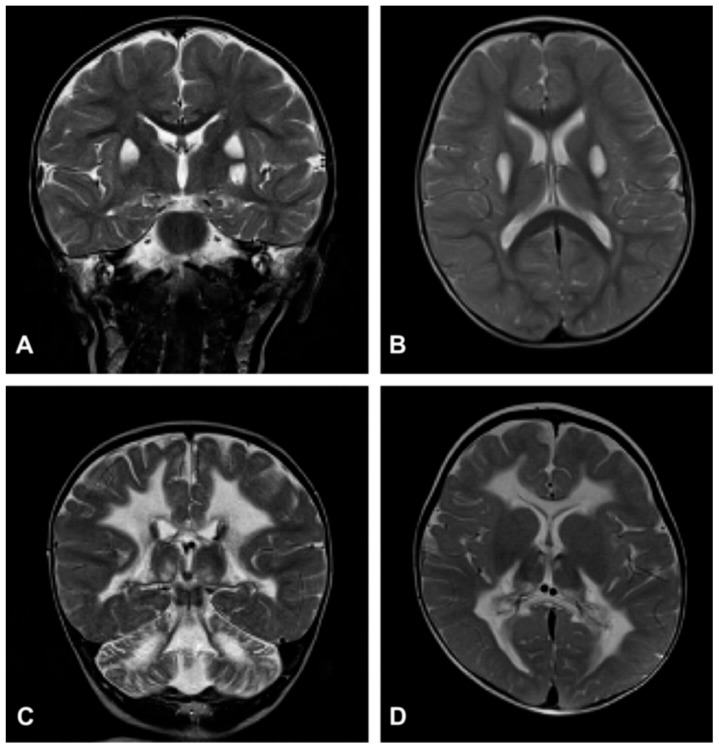
Neuroimaging in Leigh disease. (**A**,**B**) Magnetic resonance imaging (MRI) of patient presenting with Leigh phenotype with complex I deficiency due to m.10158T>C mutation in MTND3 gene: coronal (**A**) and axial (**B**) T2-weighted images show bilateral putaminal hyperintense lesions and minimal posterior periventricular white matter hyperintensity (B). (**C**,**D**) MRI of patient presenting with mitochondrial leukoencephalopathy with complex II deficiency due to mutation p.Gly169Cys of SDHAF1 gene: coronal (**C**) and axial (**D**) T2-weighted images show hyperintensity of the lobar white matter also involving the corpus callosum and the posterior arms of the internal capsule (**D**); the white matter is abnormal also in the cerebellar hemispheres (**C**). Adapted from: [115].

**Figure 4 cells-11-00637-f004:**
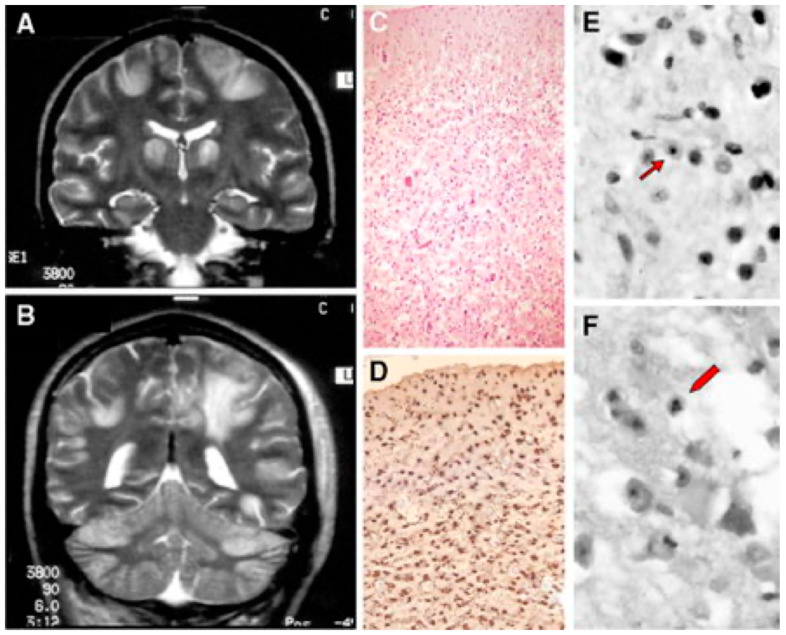
A patient affected with Alpers–Huttenlocher (AHS) diseases. (**A**,**B**) Head MRI scan (coronal plane, T2 weighed images). Bilateral focal hyperintensities are seen of the hemispheric cortex and thalami (**A**), of the white matter and cerebellar cortex (**B**). (**C**–**F**) Histological sections through cortical lesions of the same patient (HE stain; C, E, F, and GFAP; D). The pattern of the necrotizing lesions of the cortex is shown with microcavitation, vessel proliferation, neuronal loss (**C**), and the associated gliosis (**D**). Features of cell death: acute ischemic changes (arrow; **E**) and nuclear fragmentation (arrowhead; **F**) of two cortical neurons. Adapted from [132].

**Figure 5 cells-11-00637-f005:**
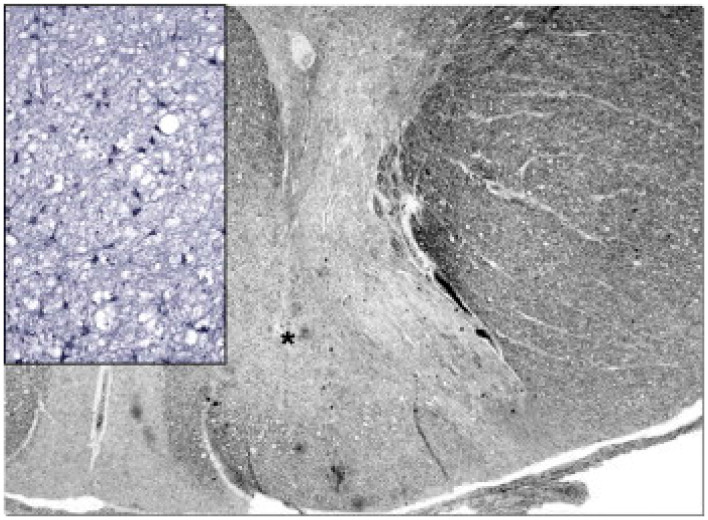
Neuropathology of the cervical spinal cord in a case of juvenile AHS. Both dorsal tracts, which carry deep sensation, are weakly stained (asterisk on the right one) and show intense gliosis (inset). Woelcke modified stain for myelin and GFAP immunohistochemistry (inset). Adapted from [114].

**Figure 6 cells-11-00637-f006:**
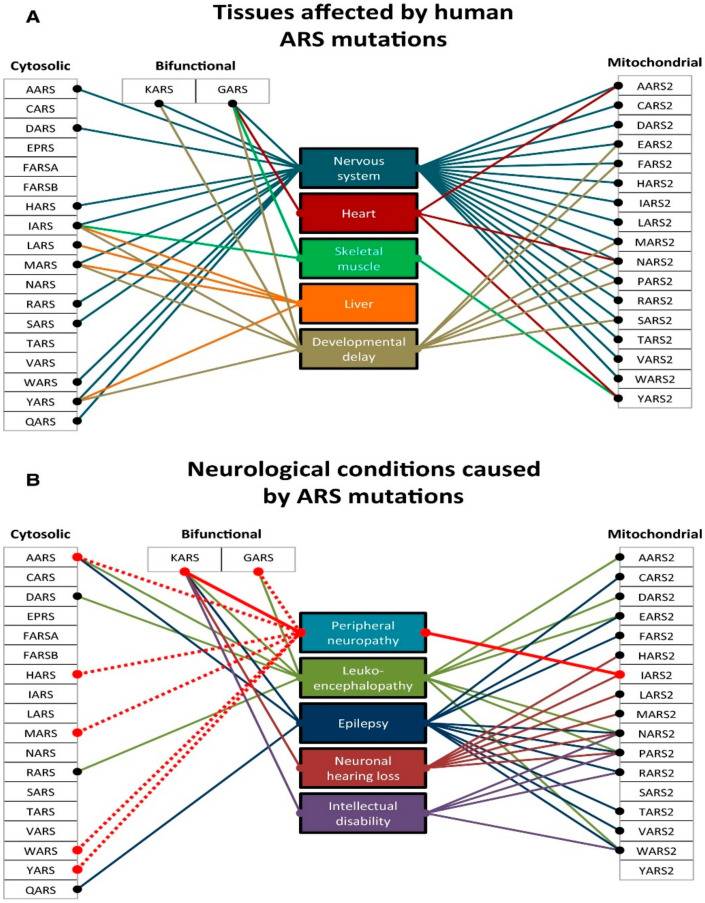
Clinical variability of diseases caused by ARSs mutations. (**A**) Tissues commonly affected by mutations in cytosolic, bifunctional, and mitochondrial ARS genes. (**B**) Common neurological presentations reported in cytosolic, bifunctional, and mitochondrial ARS genes, with peripheral neuropathy highlighted. The solid line indicates a dominant mode of inheritance, the dashed line indicates the recessive mode of inheritance. Adapted and modified from [153].

**Figure 7 cells-11-00637-f007:**
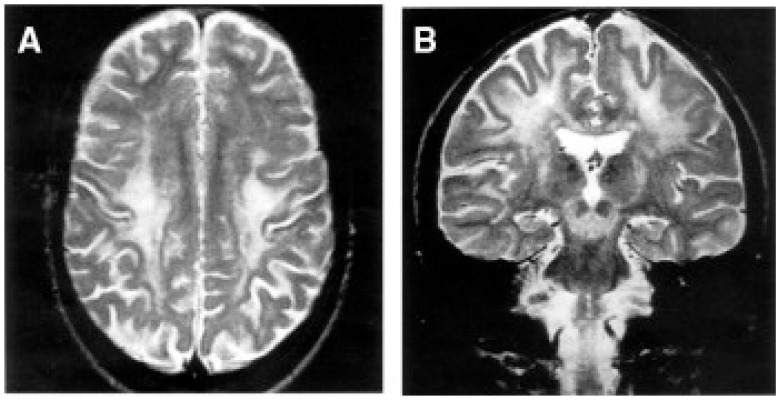
Brain MRI of a patient with KSS. An 18-year-old woman. T2-weighted transverse (**A**) and coronal (**B**) images show abnormal signals in the white matter of the centrum semiovale, especially in the perirolandic region and subcortical areas, and in the mesencephalon. From: [114].

**Figure 8 cells-11-00637-f008:**
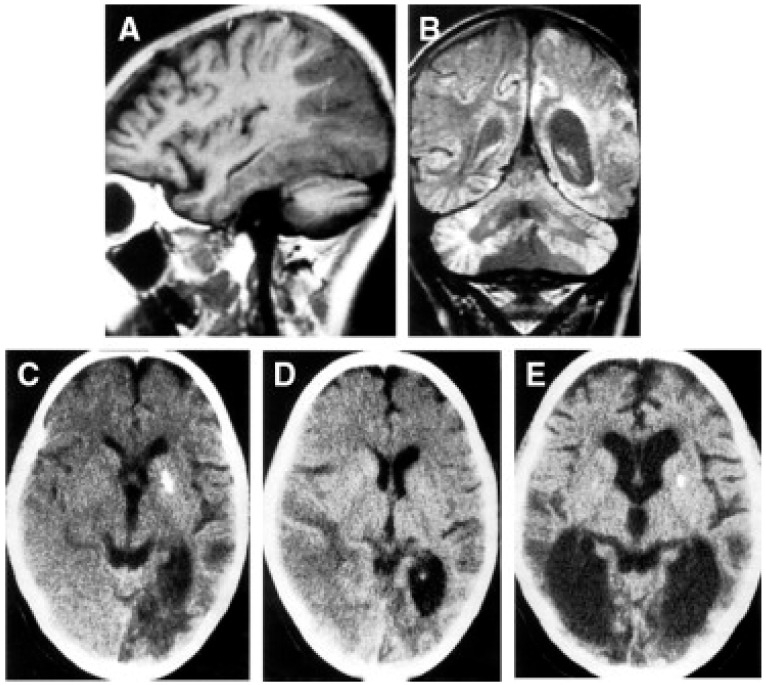
Brain MRI of a MELAS case, a 14-year-old girl. Sagittal T1-weighted section (**A**) of the left cerebral hemisphere shows a vast posterior lesion. A proton-density coronal section (**B**) shows, in addition to the temporo-occipital lesion, bilateral lesions, and atrophy of the cerebellar cortex. Serial CT scan examinations (**C**–**E**) show a reduction of the left temporal-occipital lesion (from C to D) but the appearance of a new lesion in the right temporo-occipital region (**D**). Two years later, there is marked bilateral atrophy of the posterior cerebral region (**E**). As is frequently observed in MELAS patients, as well as in other mitochondrial disorders, calcium deposits are detected in the left putamen (**C**,**E**). Adapted from [114].

**Figure 9 cells-11-00637-f009:**
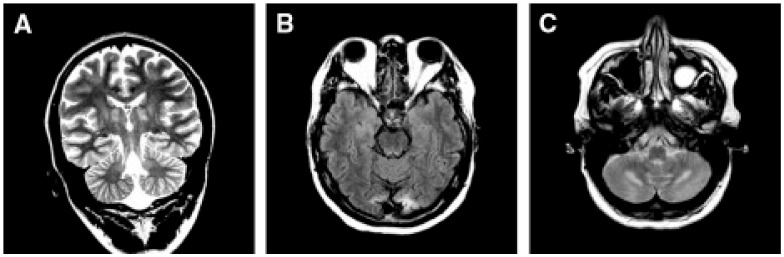
Brain MRI in SCAE. (**A**) Occipital pole lesion; (**B**) multiple lesions in cerebral cortical, subcortical, and thalamic areas; (**C**) bilateral lesions in the central cerebellar white matter. Adapted from [114].

**Table 1 cells-11-00637-t001:** Mitochondrial and nuclear encoded subunits, and (putative) assembly factors for human complex I.

Protein Name	Chromosome	Fraction	Protein	OMIM
ND1	mtDNA	Iγ	Core	516000
ND2	mtDNA	Iγ	Core	516001
ND3	mtDNA	Iγ	Core	516002
ND4	mtDNA	Iβ	Core	516003
ND4L	mtDNA	Iγ	Core	516004
ND5	mtDNA	Iβ	Core	516005
ND6	mtDNA	Iα	Core	516006
NDUFS1	2q33-q34	Iλ	Core	157655
NDUFS2	1q23	Iλ	Core	602985
NDUFS3	11p11.11	Iλ	Core	603846
NDUFS7	19p13.3	Iλ	Core	601825
NDUFS8	11q13	Iλ	Core	602141
NDUFV1	11q13	Iλ	Core	161015
NDUFV2	18p11.31-p11.2	Iλ	Core	600532
NDUFS4	5q11.1	Iλ	Accessory	602694
NDUFS5	1p34.2-p33	Iα	Accessory	603847-no dis
NDUFS6	5p15.33	Iλ	Accessory	603848
NDUFA1	Xq24	Iα	Accessory	300078
NDUFA2	5q31	Iλ	Accessory	602137
NDUFA3	19q13.42	Iα	Accessory	60383-no dis
NDUFA4	7p21.3	nd	CIV subunit	603833
NDUFA5	20p12.1	Iλ	Accessory	612360-no dis
NDUFA6	22q13.2-q13.31	Iα	Accessory	602138
NDUFA7	19p13.2	Iλ	Accessory	602139-no dis
NDUFA8	9q33.2-q34.11	Iα	Accessory	603359
NDUFA9	12p13.3	Iα	Accessory	603834
NDUFA10	2q37.3	Iα, loosely	Accessory	603835
NDUFA11	19p13.3	Iα	Accessory	612638
NDUFA12	12q22	Iλ	Accessory	614530
NDUFA13	19p13.2	Iλ	Accessory	609435
NDUFAB1	16p12.2	Iα + Iβ	Accessory	603836-no dis
NDUFB1	14q32.12	Iβ	Accessory	603837-no dis
NDUFB2	7q34	Iβ	Accessory	603838-no dis
NDUFB3	2q31.3	Iβ	Accessory	603839
NDUFB4	3q13.33	Iα + Iβ	Accessory	603840-no dis
NDUFB5	3q26.33	Iβ	Accessory	603841-no dis
NDUFB6	9p21.1	Iβ	Accessory	603322
NDUFB7	19p13.12-p13.11	Iβ	Accessory	603842-no dis
NDUFB8	10q23.2-q23.33	Iβ	Accessory	602140
NDUFB9	8q13.3	Iβ	Accessory	601445
NDUFB10	16p13.3	Iβ	Accessory	603843
NDUFB11	Xp11.23	Iβ	Accessory	300403
NDUFC1	4q28.2-q31.1	Iγ	Accessory	603844-no dis
NDUFC2	11q14.1	Iβ	Accessory	603845
NDUFV3	21q22.3	Iλ	Accessory	602184
NDUFAF1	15q11.2-q21.3		Assembly	606934-no dis
NDUFAF2	5q12.1		Assembly	609653
NDUFAF3	3p21.31		Assembly	612911
NDUFAF4	6q16.1		Assembly	611776
Ecsit	19p13.2		Assembly	608388-no dis
C20orf7	20p12.1		Assembly	612360
C8orf38	8q22.1		Assembly	612392
ACAD9	3q21.3		Assembly	611103
NUBPL	14q12		Assembly	613621
FOXRED1	11q24.2		Assembly	613622
DNAJC30	7q11.23		Assembly	618202
NDUFA7	2p22.2		Assembly?	615898
DHDPSL/C10orf65	10q24.2		Assembly?	613597
OXCT2	1p34		Assembly?	610289-no dis
OXCT1	5p13.1		Assembly?	601424
IVD	15q14-q15		Assembly?	607036
DCI	16p13.3		Assembly?	600305-no dis
MCCC2	5q12-q13		Assembly?	609014
GPAM	10q25.2		Assembly?	602395-no dis
C7orf10	7p14		Assembly?	609187
AMACR	5p13		Assembly?	604489
PHYH	10p13		Assembly	602026
LACTB	15q22.1		Assembly?	608440
LYRM5	12p12.1		electron transfer flavoprotein regulatory factor ETFRF1 no known disease. Assemby?	Not present in OMIM

The term ‘no dis’ in the OMIM column designates the absence of reported disease. Disease-causing mutations have been described in the genes encoding the complex I assembly factors and/or cell biological studies have shown the involvement of these proteins in the assembly of complex I. On the contrary, the role of the putative assembly factors (indicated with the term ‘Assembly?’) needs to be established. As indicated, NDUFA4 has unequivocally been recently attributed to complex IV (cytochrome c oxidase) [13]. Table adapted and modified from [14].

**Table 2 cells-11-00637-t002:** Mutations in nuclear genes associated with mitochondrial complex II deficiency. Adapted and modified from [21].

Mutated Gene	Molecular Role	Main Clinical Features	OMIM Number
*SDHA*	Subunit	Mitochondrial complex II deficiency; Leigh syndrome; Dilated cardiomyopathy; Paragangliomas.	600857 614165
*SDHB*	Subunit	Gastrointestinal stromal tumors; Paragangliomas; Pheochromocytomas; Mitochondrial complex II deficiency;Leukodystrophy.	185470 115310
*SDHC*	Subunit	Gastrointestinal stromal tumors; Paragangliomas.	602413 605373
*SDHD*	Subunit	Gastrointestinal stromal tumors; Paragangliomas; Pheochromocytomas; Mitochondrial complex II deficiency. Encephalomyopathy;Prenatal hypertrophic cardiomyopathy.	602690
*SDHAF1*	Assembly Factor	Mitochondrial complex II deficiency; Leukoencephalopathy.	612848
*SDHAF2*	Assembly Factor	Paragangliomas; Pheochromocytomas.	613019 601650

**Table 3 cells-11-00637-t003:** Mutations in genes associated with mitochondrial CIII deficiency. Adapted and modified from [24] under the Creative Commons Attribution (CC BY) license. The symbol * in OMIM designates a gene code.

Protein Name	Chromosome	Molecular Role	OMIM
Cytochrome b	mtDNA	Catalytic subunit	* 516020
UQCRB	8q22.1	Accessory subunit	* 191330
UQCRQ	5q31.1	Accessory subunit	* 612080
UQCRC2	16p12.2	Accessory subunit	* 191329
CYC1	8q24.3	Catalytic subunit	* 123980
TTC19	17p12	Unknown	* 613814
BCS1L	2q35	UQCRFS1 translocase	* 603647
MZM1L	5q23.3-q31.1	UQCRFS1 chaperone	* 615831
UQCC2	6p21.31	MT-CYB translational activator and chaperone	* 614461
UQCC3	11q12.3	MT-CYB chaperone	* 616097

**Table 4 cells-11-00637-t004:** Mutations associated with mitochondrial CIV deficiency. Adapted from [51].

Gene/Protein	OMIM	Function	Reported Clinical Phenotypes
*MT-CO1*	516030	Catalytic core subunit 1	LHON, AISA, ataxia, hypotonia, and epilepsy
*MT-CO2*	516040	Catalytic core subunit 2	Optic atrophy, ataxia, myopathy, lactic acidosis, and cardiomyopathy
*MT-CO3*	516050	Catalytic core subunit 3	LHON, myoglobinuria, lactic acidosis, encephalopathy, tetraparesis, and myopathy
*COX4I1*	123864	Subunit 4 isoform 1	Poor growth, dysmorphism, Fanconi anaemia, and encephalopathy
*COX4I2*	607976	Subunit 4 isoform 2	Congenital exocrine pancreatic insufficiency
*COX5A*	603773	Subunit 5A	Pulmonary arterial hypertension, lactic acidosis, and failure to thrive
*COX6A1*	602072	Subunit 6A isoform 1	CMTRID
*COX6A2*	602009	Subunit 6A isoform 2	Myopathy
*COX6B1*	124089	Subunit 6B isoform 1	Encephalomyopathy, hypotonia, growth retardation, and lactic acidosis
*COX7A1*	123995	Subunit 7A isoform 1	Failure to thrive, encephalopathy, and hypotonia
*COX7B*	300885	Subunit 7B	MLS and MIDAS
*COX8A*	123870	Subunit 8A	Pulmonary hypertension, microcephaly, anddevelopmental delay
*COXFA4*	603933	Subunit FA4	Encephalopathy, dystonia, ataxia, and lactic acidosis
*COX14*	614478	MT-CO1 stabilization	Encephalopathy, lactic acidosis, and respiratory distress
*COA3*	614775	MT-CO1 stabilization	Exercise intolerance and peripheral neuropathy
*TACO1*	612958	MT-CO1 translational activation	LD, optic atrophy, hypotonia, and tetraparesis
*COX10*	602125	Heme A biogenesis	Ataxia, hypotonia, lactic acidosis, sensorineural loss, and Leigh syndrome
*COX15*	603646	Heme A biogenesis	Cardioencephalomyopathy and LD
*COX20*	614698	MT-CO2 stabilization	Growth retardation, hypotonia, cerebellar ataxia, and lactic acidosis
*SCO1*	603644	CuA centre biogenesis	Encephalopathy, liver disease, hepatomegaly, lactic acidosis, and cardiac hypertrophy
*SCO2*	604272	CuA centre biogenesis	Encephalo–cardiomyopathy
*SURF1*	185620	Unknown	LD
*COA5*	613920	Unknown	Cardiomyopathy
*COA6*	614772	CuA centre biogenesis	Hypertrophic cardiomyopathy
*COA7*	615623	Unknown	Encephalopathy and spinocerebellar ataxia
*COA8*	616003	Unknown	Encephalopathy, cavitating dystrophy, tetraparesis, and ataxia
*PET10*0	614770	Unknown	LD and lactic acidosis
*PET117*	614771	Unknown	Neurodevelopmental regression, exercise intolerance, and lactic acidosis
*FASTKD2*	612322	mt-mRNAs stability	MELAS, brain atrophy, developmental delay, hemiparesis, and encephalomyopathy
*LRPPRC*	607544	mt-mRNAs stability	French-Canadian Leigh syndrome

**Table 5 cells-11-00637-t005:** Mutations in nuclear and mitochondrial genes associated with mitochondrial complex V deficiency. Adapted and modified from [52] under the Creative Commons Attribution (CC BY) license. * indicates a STOP codon.

Subunit or Assembly Factor	mtDNA or nDNA Mutation	Protein Mutation	Assembly	OMIM
ATP6 (a subunit)	m.8993T>G	p.Leu156Arg	Normal	516060
	m.8993T>C	p.Leu156Pro	Nd	516060
	m.9176T>G	p.Leu217Arg	Impaired	516060
	m.9176T>C	p.Leu217Pro	Impaired	516060
	m.9035T>C	p.Leu170Pro	Nd	Not present
	m.9185T>C	p.Leu220Pro	Nd	5160600
	m.9191T>C	p.Leu222Pro	Impaired (in yeast)	Not present
	m.8969G>A	p.Ser148Asn	Nd	516060
	m.8611_8612 insC	p.Leu29Profs *36	Impaired	516060
ATP6 (a subunit) and ATP8 (A6L subunit)	m.8528T>C	a p.Met1Thr + A6L p.Trp55Arg	Impaired	516060
	m.8529G>A	a p.Met1Ile + A6L p.Trp55 *	Impaired	516070
	m.8561C>G	a p.Pro12Arg + A6L p.Pro66Ala	Impaired	516060-516070
	m.8561C>T	a p.Pro12Leu + A6L p.Pro66Ser	Impaired	516060-516070
ATP5F1E (ε subunit)	c.35A>G	p.Tyr12Cys	Impaired	606153
ATP5F1A (α subunit)	c.985C>T	p.Arg329Cys	Impaired	164360
	c.962A>G	p.Tyr321Cys	Nd	164360
ATP5F1D (δ subunit)	c.245C>T	p.Pro82Leu	Impaired	603150
	c.317T>G	p.Val106Gly	Impaired	603150
ATP5MK (DAPIT subunit)	c.87+1G>C	/	Impaired	615204
ATPAF2	c.280T>A	p.Trp94Arg	Impaired	688918
TMEM70	c.317–2A>G	/	Impaired	612418

**Table 6 cells-11-00637-t006:** Pathologies due to mutations on mitochondrial aminoacyl-tRNA synthetases. Adapted from [159].

Gene	Protein	Main Phenotype	Main Organ Affected	Age At Onset
**Mt-aaRSs related to clinical manifestations exclusively in the CNS**				
*CARS2*	Mt-CysRS	Mitochondrial epileptic encephalopathy	Brain	Infancy–Childhood
*FARS2*	Mt-PheRS	Alpers encephalopathy		Infancy
*NARS2*	Mt-PheRS	Alpers’ syndrome		Infancy–Childhood
*PARS2*	Mt-AsnRS	Alpers’ syndrome		Infancy
*RARS2*	Mt-ArgRS	Pontocerebellar hypoplasia type 6 (PCH6)		Infancy
*TARS2*	Mt-ThrRS	Fatal mitochondrial encephalomyopathy		Infancy
**Leukoencephalopathies**				
*DARS2*	mt-AspRS	Leukoencephalopathy with brainstem, spinal cord involvement and lactate elevation	Brain	Childhood–Adulthood
*EARS2*	Mt-GluRS	Leukoencephalopathy with thalamus and brainstem, and lactate elevation		Infancy
*MARS2*	Mt-MetRS	Autosomal recessive spastic ataxia with leukoencephalopathy		Childhood–Adulthood
*WARS2*	Mt-TrpRS	Intellectual disability		Infancy–Adulthood
**Mt-aaRSs correlated with clinical manifestations in the CNS and other systems**				
*VARS2*	MtValRS	Fatal mitochondrial encephalocardiomyopathy	Brain and heart	Infancy
*AARS2*	Mt-AlaRS	Leukoencephalopathy with ovarian failure	Brain and ovaries	Childhood–Adulthood
*HARS2*	Mt-HisRS	Perrault syndrome	Brain and ovaries	Childhood–Adulthood
*LARS2*	Mt-LeuRS	Perrault syndrome	Brain and ovaries	Childhood–Adulthood
*IARS2*	Mt-IleRS	CAGSSS and Leigh syndrome	Brain and musculoskeletal	Infancy–Childhood
**Mr-aaRSs correlated with clinical manifestations in the CNS or other systems**				
*SARS2*		Progressive spastic paresis	Brain	Infancy
		Hyperuricemia, pulmonary hypertension, renal failure in infancy, and alkalosis (HUPRA)	Kidney	Infancy
**Mt-aaRSs correlated with clinical manifestations in a system other than the CNS**				
*AARS2*	Mt-AlaRS	Hypertrophic cardiomyopathy	Heart	Infancy
*YARS2*	Mt-TyrRS	Myopathy, lactic acidosis, andsideroblastic anaemia 2 (MLASA2)	Muscle/bone marrow	Childhood–Adulthood

## Data Availability

In this section, data supporting reported results can be found in the reported website, as well as in the references of the paper.

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
