# Peer review of "Mitochondrial Neurodegeneration"

_cells, 2022, doi:10.3390/cells11040637_

Round 1

Reviewer 1 Report

It is a nicely summarized article on mitochondrial DNA changes in neurodegenerative diseases. I sincerely appreciate authors effort. However, most part of the material in this article is published else where by authors. It is sad authors missed to cover prominent and well recognized neurodegenerative diseases, such as Parkinson's, Alzheimer's and others. I want to see separate section on these diseases.   

Author Response

Although we understand the reviewer's point, we wish to stress the fact that our work focuses on neurodegenerative diseases for which mitochondrial dysfunction has a proven pathogenetic role. We clearly stated in the text that "we will not discuss here the role of mitochondria in common neurodegenerative diseases, such as Parkinson, Alzheimer, ALS. For these conditions we refer to elsewhere works". We think that this controversial topic would deserve a paper by itself. Also, we are not clear with the comment from the reviewer that the references are not full appropriate, while he did not give any specific suggestion.

Reviewer 2 Report

This is a very comprehensive novel review with updated information about mitochondrial diseases affecting CNS.

I have only some minor comments related to the presentation:

  1. Table 3: Please adjust the style to the other tables.
  2. Table 5 (line 252): ...in nuclear 'and mitochondrial' genes...  Please add.
  3. Table 5: Please correct OMIM numbers (remove point and additional digits)
  4. Legend to Fig. 3: Please correct to mt.10158 T>C (line 489) and p.Gly169Cys (line 492).
  5. What means LSe (line 861) ?
  6. In general I would recommend to carefully check the nomenclature of gene mutations and genes

Author Response

Thank you.

Round 2

Reviewer 1 Report

all is well